# Multiple Noises in Diffusion Model for Semi-Supervised Multi-Domain Translation

**Tsiry Mayet**
*INSA Rouen Normandie, LITIS UR 4108,*
*F-76000 Rouen, France*

**Simon Bernard**
*Université Rouen Normandie, LITIS UR 4108,*
*F-76000 Rouen, France*

**Romain Hérault**
*Université Caen Normandie, ENSICAEN, CNRS, Normandie Univ, GREYC UMR 6072,*
*F-14000 Caen, France*

**Clément Chatelain**
*INSA Rouen Normandie, LITIS UR 4108,*
*F-76000 Rouen, France*

**Reviewed on OpenReview:** *https://openreview.net/forum?id=vYdT26kDYM*

## Abstract

In this work, we address the challenge of multi-domain translation, where the objective is to learn mappings between arbitrary configurations of domains within a defined set (such as $(D_1, D_2) \rightarrow D_3$, $D_2 \rightarrow (D_1, D_3)$, $D_3 \rightarrow D_1$, etc. for three domains) without the need for separate models for each specific translation configuration, enabling more efficient and flexible domain translation. We introduce Multi-Domain Diffusion (MDD), a method with dual purposes: i) reconstructing any missing views for new data objects, and ii) enabling learning in semi-supervised contexts with arbitrary supervision configurations. MDD achieves these objectives by exploiting the noise formulation of diffusion models, specifically modeling one noise level per domain. Similar to existing domain translation approaches, MDD learns the translation between any combination of domains. However, unlike prior work, our formulation inherently handles semi-supervised learning without modification by representing missing views as noise in the diffusion process. We evaluate our approach through domain translation experiments on BL3NDT, a multi-domain synthetic dataset designed for challenging semantic domain inversion, the BraTS 2020 dataset, and the CelebAMask-HQ dataset. The code for MDD and all data are publicly available[1].

## 1 Introduction

A domain is a set of tensors drawn from the same probability distribution $p(x)$, characterized by both shared features, common across related domains, and domain-specific features, that distinguish it from other domains. We define domain translation as a function $f_{S_i, S_j} : S_i \rightarrow S_j$ that projects the data representations from a set of source domains $S_i$ to a set of target domains $S_j$.

In a scenario with $L$ domains denoted by $\mathcal{D} = \{D_1, ..., D_L\}$, we aim to obtain a model performing all translations $f$ such that $f_{S_i, \bar{S}_i} : S_i \rightarrow \bar{S}_i$ with $S_i \in \mathcal{P}(\mathcal{D})$, where $\mathcal{P}(\mathcal{D})$ represents the power set of $\mathcal{D}$, and $\bar{S}_i = \mathcal{D} - S_i$ is the complement of $S_i$ within $\mathcal{D}$. Our goal is to develop a model that is not limited to a

---

[1]https://github.com/MaugrimEP/multi-domain-diffusion

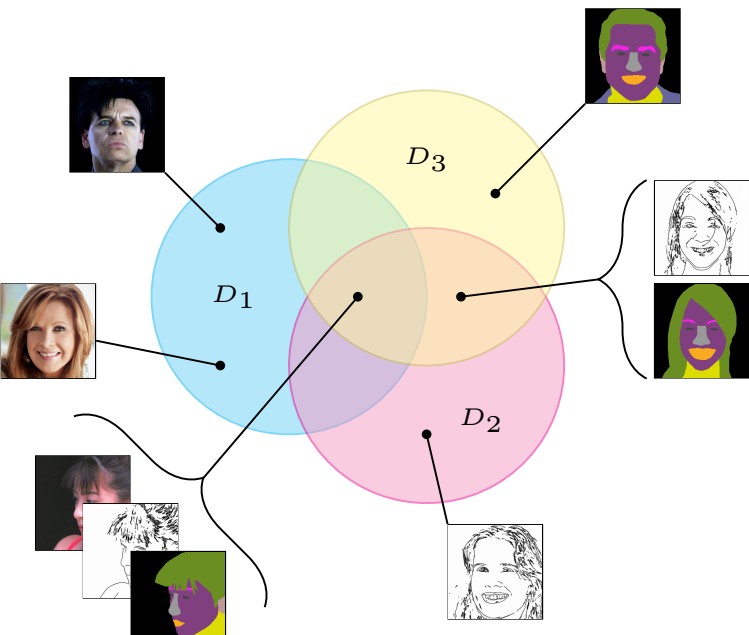

Figure 1: This example considers three domains: photographs of faces, sketches, and semantic segmentation. Supervision scenarios include full supervision (all domain samples available) and semi-supervised (some samples missing). As the number of domains increases, achieving full supervision becomes more challenging. This task can be challenging and tedious, especially when human intervention is required, such as obtaining a sketch and segmentation for each face. Semi-supervised multi-domain translation aims to reduce the data collection burden by allowing missing samples.

specific translation direction, either during training or inference. Any domain should be available to serve as a condition, while all remaining domains must be part of the generation. Given that $S_i$ can be any subset of $\mathcal{D}$, there are $2^L$ possible translation functions. Figure 1 illustrates this scenario when $L = 3$ using the CelebA-HQ (Karras et al., 2018) dataset, where $D_1$, $D_2$, and $D_3$ represent an image, a sketch, and its segmentation map, respectively. Specific instances $x^{(i)}$, such as a face $x^{(1)}$, are referred to as a view, and combinations of related views as a data point $x = [x^{(1)}, x^{(2)}, x^{(3)}]$.

Some translation configurations can be viewed as conditional generation tasks (*e.g.* face generation, where multiple faces can be considered valid given a unique sketch). In contrast, others can be viewed as regression or classification tasks (*e.g.* a unique semantic segmentation is expected given a face). This paper primarily focuses on generation and conditional generation configurations.

The MDD framework leverages the noise-removal property of diffusion models (Ho et al., 2020; Song et al., 2021; Rombach et al., 2022) and, like other frameworks, concatenate the domains in the input (Wolleb et al., 2022). In contrast to other frameworks, MDD uniquely models semi-supervised information using higher noise levels for unavailable views and further models different noise levels per domain. During training, views with higher noise levels, indicating less information, will encourage the model to rely more on less noisy views to enhance its reconstruction capabilities. This approach transforms the task from simple reconstruction to complex domain translation learning based on the joint data distribution.

**Our main contributions are summarized as:**
- We introduce the MDD framework, which incorporates multiple noise levels for each domain. This diffusion-based framework enables learning in a semi-supervised setting, allowing mapping between multiple domains.
- We investigate how the noise formulation in MDD can be used to condition the generation process on missing modalities, given a set of available views.

- We conduct a comprehensive evaluation of MDD's performance on different datasets with different modalities, using both quantitative and qualitative assessments.

## 2 Related Works

### 2.1 Domain Translation

Domain translation research has explored various generative models, including GANs (Goodfellow et al., 2014), VAEs (Kingma, 2013), normalizing flows (Rezende & Mohamed, 2015; Grover et al., 2020; Sun et al., 2019), and diffusion models (Sohl-Dickstein et al., 2015). Although approaches like Pix2Pix (Isola et al., 2017), CycleGAN (Zhu et al., 2017), and others (Mayet et al., 2023; Liu et al., 2017; Huang et al., 2018; Lee et al., 2018; 2019) have demonstrated promising results, they face limitations in multi-modal settings and exhibit reduced scalability as the number of domains increases. Moreover, both CycleGAN and StarGAN are designed for unsupervised settings, neglecting the potential benefits of supervised examples. StarGAN (Choi et al., 2018) enables translation between domain pairs using a single network for each configuration. However, it does not address multi-modal settings, where multiple modalities can be utilized as conditions simultaneously.

### 2.2 Domain Translation Using Diffusion Models

Diffusion models offer various conditional generation paradigms. These methods can be broadly categorized into two groups: those operating in the original high-dimensional pixel space and those operating in the latent space. Models working in pixel space project the condition onto the target manifold of a pre-trained model using forward diffusion on the condition (Li et al., 2023; Meng et al., 2022a). These approaches leverage the fact that, for specific translation tasks, condition and generation domain can be visually close (*e.g.* CBCT to CT (Li et al., 2023)). Methods operating in the pixel space face several limitations, including the need to have conditions and targets close together in the input space, not allowing multiple conditions, and requiring a careful balance between condition fidelity and generation realism (Meng et al., 2022a). Models working in latent space use a similar approach. They embed the condition into the latent space of the target domain, allowing the use of a pre-trained diffusion model (Wang et al., 2022; Ramesh et al., 2022; Lin et al., 2023). However, they do not address the issue of multiple conditions and target domains. Recent work on guided diffusion (Dhariwal & Nichol, 2021; Ho & Salimans, 2021; Wang et al., 2023; Cross-Zamirski et al., 2023) has explored ways to enhance adherence to condition semantics, but shares similar limitations in multi-modal settings.

### 2.3 Conditional Diffusion Models Using Concatenation

Recent approaches have attempted to address the use of multiple conditions or targets simultaneously through their concatenation (Xie et al., 2024; Cross-Zamirski et al., 2023; Saharia et al., 2022a; Lyu & Wang, 2022; Saharia et al., 2022b). However, they primarily focus on one-way translation with fixed domains. They do not address the defined multi-domain translation setting, where any domain can serve as input or output during generation. Two main approaches have emerged to overcome these limitations: noisy condition and clean condition methods. **Noisy condition:** To allow a unified framework without a fixed configuration of condition and target domains, this class of approaches (Lugmayr et al., 2022; Sasaki et al., 2021; Mariani et al., 2024) introduces the concept of adding noise to the condition. During training, the condition is degraded using the same forward diffusion process, enabling the model to learn to reverse the diffusion process for all domains using the joint data distribution. For example, RePaint (Lugmayr et al., 2022) applies a matching noise level between condition and generation and designs a jumping mechanism to maintain generation faithful to the condition while significantly increasing the generation time.

**Multi-Source Diffusion Models for Simultaneous Music Generation and Separation (MSDM)** MSDM (Mariani et al., 2024) presents an innovative approach utilizing noisy conditions. The method proposes applying an equivalent amount of noise to both the condition domain and the generated target domain, enabling multi-domain translation learning in a supervised setting. While this formulation has demonstrated success in music generation, related applications (Lugmayr et al., 2022; Meng et al., 2022a;

Chung et al., 2022; Corneanu et al., 2024; Mayet et al., 2025) have highlighted the limitations of noisy conditions and the necessity for additional mechanisms to synchronize the condition and target domains.

These limitations motivate the exploration of alternative approaches, such as UMM-CSGM, which investigates the use of clean conditioning.

**Clean Condition:** In contrast, clean condition approaches (Xie et al., 2024; Cross-Zamirski et al., 2023; Saharia et al., 2022a; Lyu & Wang, 2022; Saharia et al., 2022b) keep the condition clean during both training and generation. This scheme allows for learning a one-way translation conditioned on a specific domain and produces successful results. However, it falls short in a multi-domain translation setting, where any domain can become an input or an output at generation time. **Unified Multi-Modal Conditional Score-based Generative Model (UMM-CSGM):** Recently, UMM-CSGM (Meng et al., 2022b) has alleviated the problems of clean and noisy conditioning. It aims to learn a multi-domain medical image completion task using a multi-in multi-out conditional score network. UMM-CSGM adopts the concept of clean conditioning and incorporates a code to indicate the conditional configuration by partitioning the domain into noisy targets or clean conditions.

While this formulation enables multi-domain translation in a fully supervised context, our work adopts a different approach by embedding information about conditions and target domains directly into the noise level modeling during training. Our proposed method inherently facilitates the configuration of missing modalities and aims to overcome the limitations of previous methods in addressing flexible multi-domain translation scenarios.

## 3 Multi-Domain Diffusion (MDD) Method

### 3.1 Diffusion Model

Diffusion models learn a data distribution from a training dataset by inverting a noising process. During training, the forward diffusion process transforms a data point $x_0$ into Gaussian noise $x_T \sim \mathcal{N}(0, \mathbf{I})$ in $T$ steps by creating a series of latent variables $x_1, ..., x_T$ using the following equation

$$q(x_t|x_{t-1}) = \mathcal{N}(x_t; \sqrt{1 - \beta_t}x_{t-1}, \beta_t\mathbf{I}) \tag{1}$$

Where $\beta_t$ is the defined variance schedule. With $\alpha_t = 1 - \beta_t$, $\bar{\alpha}_t = \prod_{i=1}^{t} \alpha_i$, and $\epsilon \sim \mathcal{N}(0, \mathbf{I})$, $x_t$ can be marginalized at a step $t$ from $x_0$ using the reparametrization trick

$$x_t = \sqrt{\bar{\alpha}_t}x_0 + \sqrt{1 - \bar{\alpha}_t}\epsilon. \tag{2}$$

The reverse denoising process $p_\theta(x_{t-1}|x_t, t)$ allows generation from the data distribution by first sampling from $x_T \sim \mathcal{N}(0, \mathbf{I})$ and iteratively reducing the noise in the sequence $x_T, ..., x_0$. The model $\epsilon_\theta(x_t, t)$ is trained to predict the added noise $\epsilon$ to produce the sample $x_t$ at time step $t$ using mean square error (MSE):

$$\mathcal{L} = \mathbb{E}_{\epsilon \sim \mathcal{N}(0, \mathbf{I}), x_0, t} \|\epsilon_\theta(\sqrt{\bar{\alpha}_t}x_0 + \sqrt{1 - \bar{\alpha}_t}\epsilon, t) - \epsilon\|_2^2. \tag{3}$$

### 3.2 Existing Issues With Noisy Conditional Diffusion Models

We focus on noisy conditional diffusion models that concatenate different modalities as input to $\epsilon_\theta$, as detailed in Sec. 2.3. These models face inherent challenges due to the shared noise level $t$ during forward and backward diffusion processes across all domains. The fundamental issue arises from the discrepancy between noise levels of available and unavailable views during training and inference. During training, unavailable views are replaced with pure noise, while other domains contain varying levels of noise. Consequently, $t$ no longer accurately represents the true noise distribution across domains.

This discrepancy propagates to the backward diffusion process during inference (illustrated in Figs. 2 and 3), where the conditional inputs must be degraded to match the target domain's noise level (as shown in the initial steps in (a) Fig. 2, where the sketch and segmentation mask contain significant noise). When the timestep approaches $T$, the condition becomes extremely noisy, retaining minimal semantic information. This

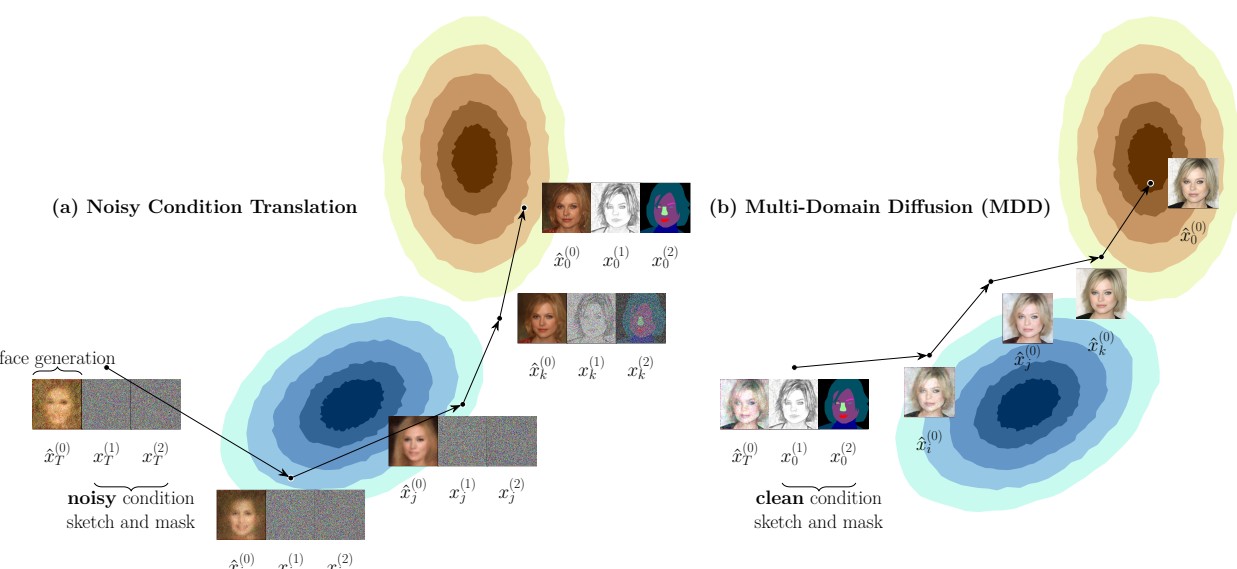

Figure 2: **Schematic illustration of Noisy Conditional Diffusion Models vs MDD, for the same timesteps** on CelebAMask-HQ for face generation given sketch and mask conditions. The generation proceeds from left to right, where $\hat{x}_I$ represents the model's ($\epsilon_\theta$) prediction at noise level $I \in [0, T]$. **(a)** MSDM$^\dagger$ (Mariani et al., 2024) employs noisy conditions. At each timestep, the model receives as input both the current generation with reinjected noise and the noisy condition. Initially, the highly noisy condition leads the model to predict blurred images and converge toward a data manifold that may not preserve the conditional features (represented by the blue). As the generation progresses and noise in the condition decreases, the model gains access to the underlying conditional information and corrects its trajectory, moving toward the appropriate data manifold (represented by the red area). The diffusion model must continuously adjust its trajectory to accommodate new information that becomes available as noise decreases. **(b)** Unlike noisy conditional models, MDD utilizes clean conditions throughout the generation process (conditional domains are shown only at the initial timestep for clarity, though they are utilized at each step). This enables the condition to guide the generation from the initial diffusion steps, resulting in intermediate generated images that are more strongly influenced by the conditional information. Zoom in for better details.

degradation causes the generation of the target domain to deviate from the intended semantics, resulting in the generation traversing different modes as it attempts to align with the gradually revealed conditional information as noise levels decrease. This phenomenon is demonstrated in Fig. 3, where the noisy condition produces inconsistent outputs, exhibiting variations in key facial features such as gender, hair color, and hair length. RePaint (Lugmayr et al., 2022) encounters a similar issue, which it addresses by implementing a jumping mechanism. This solution involves looping through the same generation steps multiple times to increase the semantic consistency, but at the cost of increased time and computational complexity.

In contrast, MDD's approach of modeling distinct timesteps per domain enables the preservation of clean conditional inputs during inference (illustrated in Fig. 2, where the conditions remain noise-free). Consequently, the generation process maintains greater consistency and is more effectively guided by the conditional information throughout the diffusion process, as demonstrated in Fig. 3, where the intermediate steps exhibit stability and manifest the final image features from the early stages of generation. This allows the diffusion model to focus on refining the sample rather than correcting its trajectory.

### 3.3 Noise Modeling for Semi-Supervised Multi-Domain Translation

To address these interconnected problems, MDD introduces an augmented forward and backward diffusion process. It employs a vector $\mathcal{T}$ of size $L$, assigning a separate $t^{(i)}$ for each of the $L$ domains. This approach

Conditions

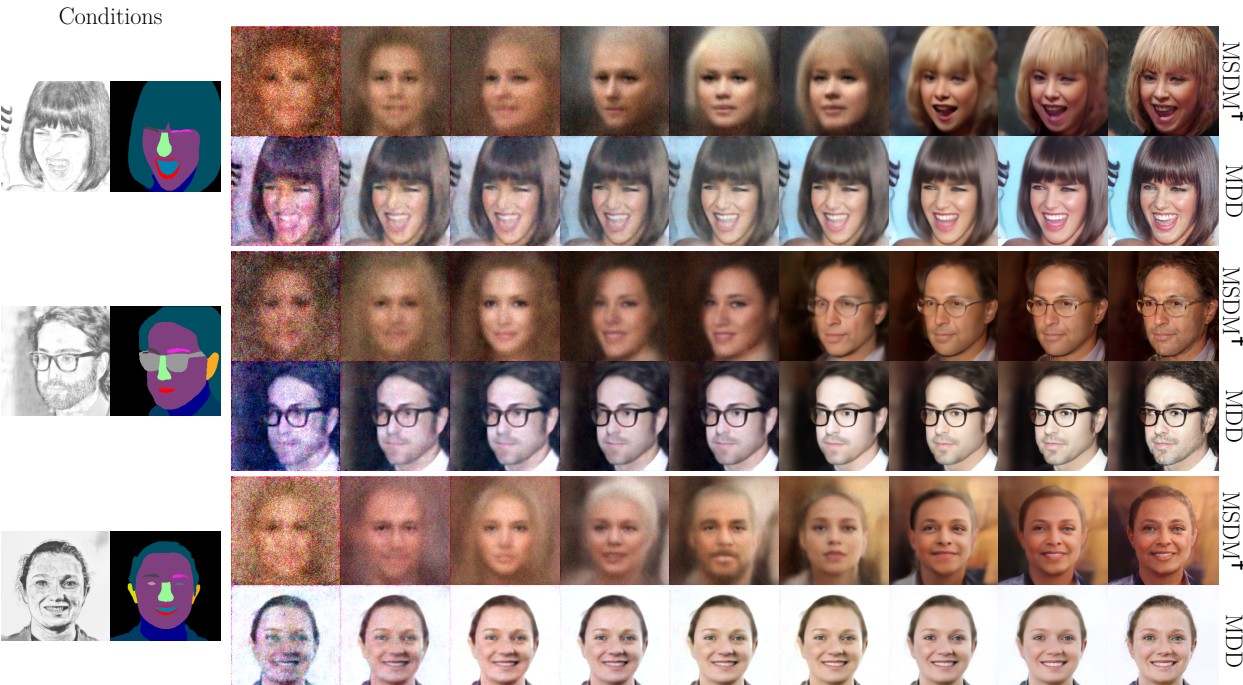

Figure 3: Illustration comparing **(top) noisy condition with MSDM$^\dagger$ (Mariani et al., 2024)** and **(bottom) clean condition with MDD** generation at equivalent timesteps on CelebAMask-HQ for face translation, conditioned on sketch and semantic segmentation inputs. The noisy condition approach exhibits mode switching during generation, resulting in inconsistent facial attributes such as gender, hair color, and length across timesteps. This occurs as the noisy condition model must adjust its trajectory to accommodate emerging features from the condition, as illustrated in Fig. 2. In contrast, MDD maintains consistency throughout the generation process, leading to higher-quality domain translation with coherent feature preservation.

allows for more accurate modeling of noise levels across different domains, removing the discrepancy between noise levels in training and issues of semantic deviation during inference.

The goal of MDD is to remove constraints on predefined conditional domains for semi-supervised domain translation. During training, $x$ represents the ground truth data point, where $x \odot m$ denotes the available supervised views and $x \odot (1 - m)$ denotes the unavailable views, and $\odot$ denotes the Hadamard product - $(A \odot B)_{i,j} = (A_{i,j} \times B_{i,j})$. The binary supervision mask $m$ indicates the presence or absence of domain supervision. For instance, in Fig. 4, the mask values are $m = [1, 0, 1]$, indicating that the second domain is missing during training. During generation, $x \odot m$ represents the conditional views, while $x \odot (1 - m)$ represents the views to be generated. This is illustrated in Fig. 5, where the mask values are $m = [0, 1, 1]$, indicating that the first domain is being generated from the second and third domains.

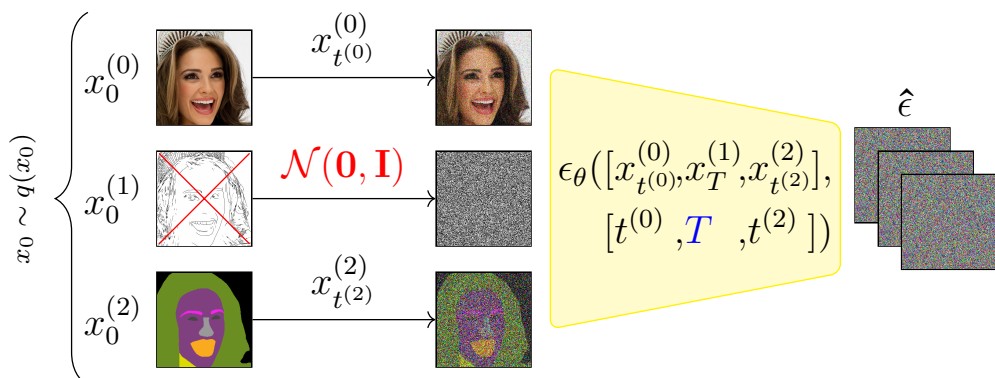

Figure 4: **Training procedure of MDD.** This example illustrates the training process across three domains: $D_0$ (face domain), $D_1$ (sketch domain), and $D_2$ (semantic face segmentation). For a given data point $x$ where $D_1$ (sketch) is missing, we substitute it with Gaussian noise $\mathcal{N}(\mathbf{0}, \mathbf{I})$ and set its corresponding time step $t^{(1)}$ to $T$. In this example, the supervision mask is defined as $m_{\mathrm{sup}} = [1, 0, 1]$, and the noise levels are initially sampled as $\mathcal{T} = [100, 500, 200]$ (Eq. (4) or Algorithm 1, Line 5). After applying Eq. (6) (Algorithm 1, Line 8), these values become $\mathcal{T} = [100, T, 200]$.

---

**Algorithm 1** MDD training process

---

1: **repeat**
2:    $x_0 \sim q(x_0)$
3:    $m_{\mathrm{sup}}$ the supervision mask
4:    $m_{\mathrm{c}} \in \{0, 1\}^L$ a random conditional mask
5:    $\mathcal{T}_{\mathrm{sup}} \sim \mathrm{Uniform}(\{1, ..., T\})^L$
6:    $\epsilon \sim \mathcal{N}(\mathbf{0}, \mathbf{I})$
7:    $\mathbf{z} \sim \mathcal{N}(\mathbf{0}, \mathbf{I})$
8:    $\mathcal{T} = T \odot (1 - m_{\mathrm{sup}}) + \mathcal{T}_{\mathrm{sup}} \odot m_{\mathrm{sup}}$
9:    $\mathcal{T} = 0 \odot (1 - m_{\mathrm{c}}) + \mathcal{T} \odot m_{\mathrm{c}}$
10:   $x_{\mathcal{T}} = \sqrt{\bar{\alpha}_{\mathcal{T}}} x_0 + \sqrt{1 - \bar{\alpha}_{\mathcal{T}}} \epsilon$
11:   $x_{\mathcal{T}} = \mathbf{z} \odot (1 - m_{\mathrm{sup}}) + x_{\mathcal{T}} \odot m_{\mathrm{sup}}$
12:   Training with loss: $\mathcal{L} = \|\epsilon - \epsilon_\theta(x_{\mathcal{T}}, \mathcal{T})\|_2^2$
13: **until** convergence

---

During training, different noise levels are sampled for each domain:

$$\mathcal{T}_{\mathrm{sup}} \sim \mathrm{Uniform}(\{1, ..., T\})^L \tag{4}$$

encouraging learning the joint data distribution by using less noisy domains to reconstruct noisier ones. In semi-supervised settings, missing domains are handled by replacing their views with Gaussian noise and masking their corresponding time steps in $\mathcal{T}$, where $t$ is replaced with $T$:

$$x_{\mathcal{T}} = \mathbf{z} \odot (1 - m_{\mathrm{sup}}) + x_{\mathcal{T}} \odot m_{\mathrm{sup}} \text{ , with } \mathbf{z} \sim \mathcal{N}(\mathbf{0}, \mathbf{I}) \tag{5}$$

$$\mathcal{T} = T \odot (1 - m_{\mathrm{sup}}) + \mathcal{T}_{\mathrm{sup}} \odot m_{\mathrm{sup}} \tag{6}$$

The forward diffusion process is applied independently for each view according to $\mathcal{T}$:

$$x_{\mathcal{T}} = \sqrt{\bar{\alpha}_{\mathcal{T}}} x_0 + \sqrt{1 - \bar{\alpha}_{\mathcal{T}}} \epsilon \tag{7}$$

The loss is computed only on available domains. The training process is illustrated in Fig. 4 using three domains from the CelebAMask-HQ dataset with sketches. Missing samples are replaced with Gaussian noise, and their corresponding time steps $t^{(i)}$ are set to $T$. Meanwhile, the supervised domains receive independently sampled noise levels.

Unless specified otherwise, in addition to Eq. (6), MDD training includes an additional step where a subset of domains is randomly selected as clean, with their associated $t$ set to 0 (*cf.* Algorithm 1, line 9).

---

**Algorithm 2** MDD generation process

---
1: $x_T \sim \mathcal{N}(\mathbf{0}, \mathbf{I})$
2: **for** $t = T, ..., 1$ **do**
3:      $\epsilon \sim \mathcal{N}(0, \mathbf{I})$
4:      $z \sim \mathcal{N}(0, \mathbf{I})$ if $t > 1$, else $z = 0$
5:      $x_{\mathrm{cond},\phi(t)} = \sqrt{\bar{\alpha}_{\phi(t)}} x_{\mathrm{cond}} + \sqrt{1 - \bar{\alpha}_{\phi(t)}} \epsilon$
6:      $\mathcal{T} = t \odot (1 - m) + \phi(t) \odot m$
7:      $x_{\mathcal{T}} = x_t \odot (1 - m) + x_{\mathrm{cond},\phi(t)} \odot m$
8:      $x_{t-1} = \frac{1}{\sqrt{\bar{\alpha}_{\mathcal{T}}}} \left( x_{\mathcal{T}} - \frac{\beta_{\mathcal{T}}}{\sqrt{1 - \bar{\alpha}_{\mathcal{T}}}} \epsilon_\theta(x_{\mathcal{T}}, \mathcal{T}) \right) + \sigma_{\mathcal{T}} z$
9: **end for**
10: **return** $x_0$

---

This modification allows the network to recognize unavailable views as having a maximum noise level (modeled by $T$) and replaced by noise. During training, the model sees views with different noise levels, encouraging it to exploit more inter-domain dependencies for reconstruction. The training procedure is detailed in Algorithm 1. This modification also allows MDD to take input with different noise levels for direct generation using a clean condition, differing from other work where conditions must have the same noise level as the target domain or where condition and target domain are predefined. The new generation process is described in Algorithm 2 and illustrated in Fig. 5, where $\phi(t)$ controls the amount of information in the condition during backward diffusion. Unless specified otherwise, $\phi(t)$ is set to $\phi(t) = 0$ in our results. We investigate different functions for $\phi(t)$ in Appendix B.1. Intuitively, if $\phi(t)$ results in a function producing a higher noise level, the generation will drift further away, allowing it to produce more diverse results at the cost of less fidelity to the condition.

Similar to previous works on image translation (Meng et al., 2022a), different $\phi$ strategies enable achieving a trade-off between image diversity and faithfulness to the conditioning signal. Applying higher noise levels to the condition allows the generation process to deviate from a specific unique output, producing more diverse images in generative tasks where multiple distinct outputs are desired. Conversely, for discriminative tasks such as semantic segmentation, it is preferable to maintain a low noise level on the condition, ideally 0, to achieve higher precision in the predicted outputs.

The MDD approach addresses the multi-domain semi-supervised translation task by modeling an independent noise level for each domain. The design of sampling independent noise levels aims to transform the reconstruction task of noisy condition models into a translation task, wherein cleaner views are utilized to predict noisier views. This formulation inherently accommodates the semi-supervised setting by representing missing views as noise. As all views are concatenated in the model input and the model simultaneously generates all modalities, it eliminates the need for duplications across various input or output configurations.

## 4 Experiments

### 4.1 Datasets

We validate MDD on three datasets, each containing more than two domains: 1) our proposed BL3NDT synthetic dataset, 2) the BraTS 2020 (Menze, 2014; Spyridon Bakas et al., 2017; et al., 2019; 2017) medical dataset with missing modality completion task, and 3) CelebAMask-HQ (Lee et al., 2020) dataset of face photos and masks, augmented with generated sketches.

Each dataset is evaluated under varying levels of supervision. Model performance is assessed by removing a specific number of views from a data point and subsequently regenerating them.

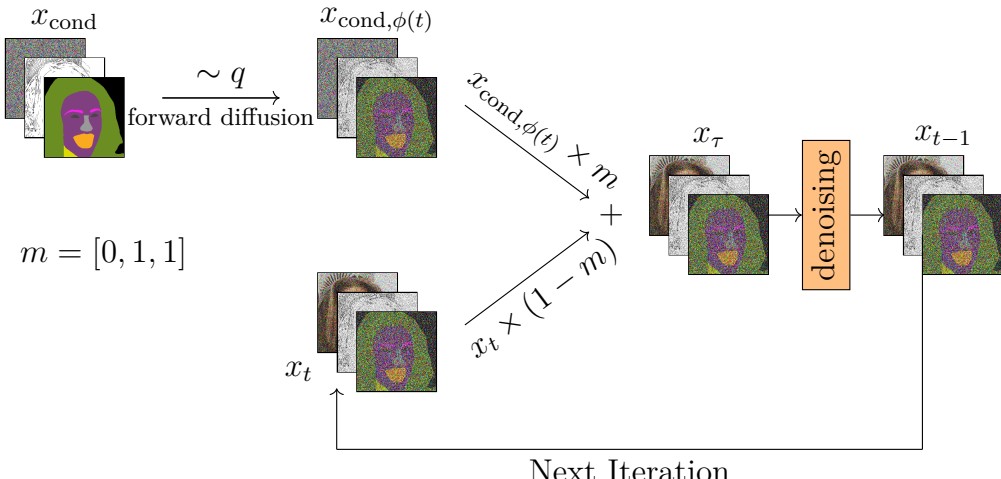

Figure 5: **Conditional generation procedure of MDD.** This example illustrates the training process across three domains: $D_0$ (face domain), $D_1$ (sketch domain), and $D_2$ (semantic face segmentation). The first domain (face) is missing and is considered a target domain to generate, while the other two domains (sketch and semantic face segmentation) are available and are considered a condition. $m$ is the mask indicating missing and available domains, and $\phi(t)$ controls the noise added to the condition.

### 4.1.1 BL3NDT Synthetic Dataset

The Blender 3 Domain Translation (BL3NDT) dataset provides a testbed for multi-domain translation frameworks with deterministic mapping. Generated using the open-source 3D software Blender[2], it consists of 64x64 image triplets across three domains (cube, pyramid, iscosphere), each containing domain-specific and common features (Fig. 6). Each domain represents an object (cube, pyramid, or icosphere) placed before two walls and a floor, with controllable viewing angles. Generation parameters include object type, 3D position, camera angle, object color, floor color, and wall colors. Some parameters are common across domains, while others (position, camera angle, object color) exhibit semantic inversion between domains (Tab. 1). By swapping generative parameters between domains, pixel-to-pixel mapping is eliminated, compelling the model to learn underlying generative parameters and increasing task difficulty. The dataset comprises 40,500 randomly generated image triplets.

We study the BL3NDT setting with different amounts of supervised data. A percentage of the dataset, denoted as N%, is considered supervised data points, with the remaining (100-N)% divided equally between pairs of domains, with $N \in \{100, 70, 10, 0\}$. For example, if $N = 40\%$, 40% of the data is fully supervised, 20% is (cube, pyramid), 20% is (pyramid, icosphere), and 20% is (cube, icosphere). In addition, the *bridge setting* divides the dataset into 50% (cube, pyramid) and 50% (pyramid, icosphere) pairs, by removing the icosphere from half the data and removing the cube from the other half of the data. We call it *bridge translation* as there is never a cube and a corresponding icosphere together. Therefore, producing an icosphere from a cube requires using a bridge domain: the pyramids. Generation is evaluated using the Mean Average Error (MAE).

Table 1: Parameters controlling BL3NDT dataset generation. Position, Camera angle, and Object color semantics are swapped between domains, while floor and wall colors are shared.

|  | Position | Camera angle | Object color | Floor color | Wall 1 color | Wall 2 color |
|---|---|---|---|---|---|---|
| Cube | p1, p2, p3 | $\alpha$ | $\alpha$, color1, color2 | r1, g1, b1 | r2, g2, b2 | r3, g3, b3 |
| Pyramid | p2, p3, p1 | $1 - \alpha$ | color1, color2, $\alpha$ | r1, g1, b1 | r2, g2, b2 | r3, g3, b3 |
| Icosphere | p3, p1, p2 | $(\alpha + 0.5)\%1$ | color2, $\alpha$, color1 | r1, g1, b1 | r2, g2, b2 | r3, g3, b3 |

---

[2]https://www.blender.org/

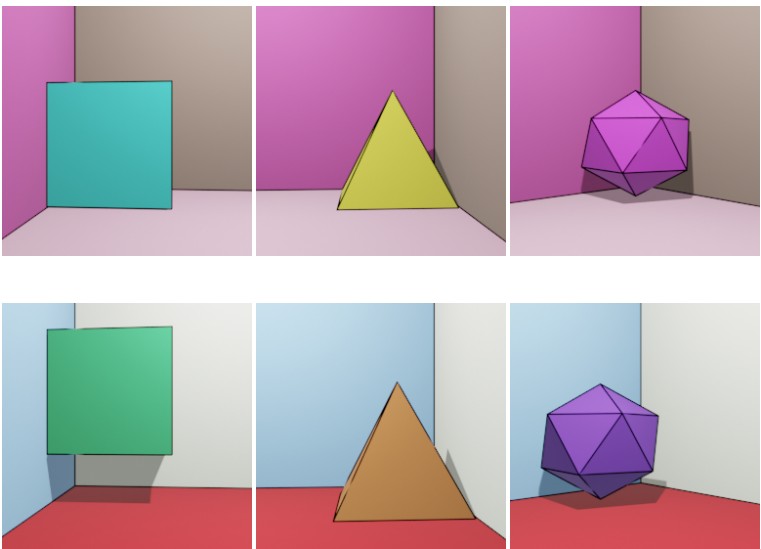

Figure 6: Two data points from BL3NDT dataset which provides a synthetic domain translation setup.

### 4.1.2 CelebAMask-HQ Dataset Augmented With Sketches

Evaluations are conducted on the CelebAMask-HQ dataset at 256x256 resolution. To obtain a setting with more than two domains, each face is augmented with a corresponding sketch computed using a pre-trained model (Chen et al., 2018).

The evaluation focuses on face-conditioned generation, as MDD is designed for generation rather than discriminative tasks (*e.g.* classification, regression, segmentation *etc.*).

The same supervision settings as BL3NDT is used with N=100% and N=0%. The quality of face generation is evaluated using established metrics from image-to-image translation, the Learned Perceptual Image Patch Similarity (Zhang et al., 2018)(LPIPS) and the Structural Similarity Index Measure (Wang et al., 2004) (SSIM).

### 4.1.3 BraTS 2020 Medical Dataset

Evaluations are conducted on the BraTS 2020 (Menze, 2014; Spyridon Bakas et al., 2017; et al., 2019; 2017) dataset comprising four MRI modalities (FLAIR, T1, T1ce, and T2), and 3D tumor segmentations. We use 2D slices (256x256) with linear normalization [0,1] and binary semantic segmentation.

As for CelebAMask-HQ, the evaluation focuses on scans generation conditioned on other modalities, since MDD is designed for generation rather than discriminative tasks.

For BraTS 2020, we evaluate three supervision settings: P100% (fully supervised), P80% (80% probability of keeping each view $x_j^{(i)}$), and P50% (50% probability of keeping each view $x_j^{(i)}$), and a minimum of two views are retained for each data point. Removing random scans simulates a realistic scenario where not all patients have complete scan sets. In the P50% setting, only 3% of the data retain all five modalities, 16% miss one modality, and 81% lack two or more modalities. This heterogeneous setting presents a more complex and realistic challenge compared to the BL3NDT dataset. Missing modality completion on the BraTS 2020 dataset is evaluated using the standard metrics for this task (Meng et al., 2022b; Li et al., 2023; Xie et al., 2024): Peak Signal-to-Noise Ratio (PSNR) and Structural Similarity Index Measure (Wang et al., 2004) (SSIM). For segmentation generation, the Jaccard Index (Jaccard) is reported.

## 4.2 Implemented Models

**Comparison with State-of-the-Art Methods (sota)** For evaluation, MDD is compared with two multi-domain translation paradigms, clean and noisy conditions, which do not specify a condition domain. For the clean condition paradigm, UMM-CSGM (Meng et al., 2022b) training scheme is used as described in Sec. 2.3. The MSDM (Mariani et al., 2024) training scheme is used for the noisy condition paradigm, which has shown competitive quantitative results in music generation and separation. UMM-CSGM uses binary vectors for flexible conditions and target domain definitions, applying forward diffusion only to the target domain and keeping the condition domain clean. MSDM applies the forward diffusion process to all domains during training, bringing them to the same noise level. In MSDM, conditional generation is performed by applying noise to the condition to match the noise level in the target domains. Since MSDM was initially designed for music generation, we kept only the noise scheduling for each domain and referred to it as MSDM$^{\dagger}$. We refer to these models as 'sota' in the different results tables. In a semi-supervised setting, 'sota' methods are only trained on the supervised examples, ignoring semi-supervised data.

**Comparison with Uni-Directional Domain Translation Methods** In addition to multi-domain methods, we conduct comparisons with state-of-the-art diffusion translation models that are restricted to a specific translation direction ($f_{S_i,\bar{S}_i} : S_i \rightarrow \bar{S}_i$) using ControlNet (Zhang et al., 2023b) in Appendix B.6. While ControlNet does not strictly fit the introduced multi-domain translation setting, as it operates only on fixed, predefined translation directions, we include these comparisons for the sake of completeness.

**Adaptation of State-of-the-Art Methods to Semi-Supervised (ad.)** To our knowledge, only UMM-CSGM, MSDM, and the like, are capable of addressing the defined multi-modal domain translation setting. To evaluate performance in both supervised and semi-supervised scenarios, we note that UMM-CSGM, MSDM, and ControlNet do not inherently support semi-supervised training. Therefore, we propose modified variants of each model (UMM[N], MSDM$^{\dagger}$[N], and ControlNet[N]) that handle missing views by substituting them with noise. The different results tables refer to these adapted models as 'ad.'.

**Multi-Domain Diffusion Ablations (ours)** To extensively test MDD capabilities, further ablation is provided by removing the contribution of Eq. (4), MDD training scheme is adapted to noisy conditions (named MDD[NOISY]). $\mathcal{T}$ is set as $\mathcal{T} = [t,t,t]$ during training. During generation, missing domains are replaced with noise and their $t$ with $T$ as specified in Eq. (6). Ablation of the additional step in Algorithm 1, line 9 is also provided (named MDD[RAND]). The different results tables refer to these adapted models as 'ours' and are analyzed in Appendix B.2.

Additional training details are provided in Appendix A.

## 4.3 BL3NDT Results

In this section, we demonstrate that MDD effectively performs domain translation on datasets without strong pixel-to-pixel correspondence between domains. We examine three distinct supervision settings: (1) the fully-supervised setting, where all data views are available; (2) the semi-supervised setting, where some views are missing from data points; and (3) the bridge data setting, where certain domains are never simultaneously present in the training samples.

In Tab. 2, MDD is compared to the different baselines. For all amount of supervision considered, MDD outperforms every baseline. The lower the supervision (lower $N$), the more MDD outperforms other baselines.

**Supervised performances:** For $N = 100\%$, UMM-CSGM, UMM[N], and MDD exhibit comparable low MAE errors. In contrast, MSDM$^{\dagger}$ and MSDM$^{\dagger}$[N] demonstrate notably higher MAE error. The similarity in results for UMM-CSGM, UMM[N], and MDD is expected, as they all utilize a clean condition during the generation process, and the full supervision allows learning the translation task. The higher MAE errors observed in MSDM$^{\dagger}$ and MSDM$^{\dagger}$[N] can be attributed to their noisy condition: the condition is degraded to the same level as the generation. This creates a disharmony between condition and generation, as the model cannot leverage the condition during earlier steps. This shows how using a clean condition guides the generation in the right direction from the start of the diffusion process, resulting in more faithful generations.

Table 2: MAE↓ error (values are given in e-1 order) for different amounts of supervision on the BL3NDT dataset for cube→(pyramid, icosphere) translation. 'sota' refers to the original state-of-the-art methods without modifications, while 'ad.' refers to our adapted versions of these methods modified to handle semi-supervised scenarios.

| | BL3NDT MAE↓ | 100% | 70% | 40% | 10% | 0% | Bridge |
|---|---|---|---|---|---|---|---|
| sota | MSDM† | 1.962 | 2.028 | 2.186 | 2.925 | - | - |
| | UMM-CSGM | 0.359 | 0.503 | 0.852 | 4.601 | - | - |
| ad. | MSDM†[N] | 1.976 | 1.944 | 1.971 | 2.060 | 2.321 | 3.036 |
| | UMM[N] | 0.353 | 0.422 | 0.443 | 0.544 | 3.324 | 3.444 |
| | MDD | **0.316** | **0.338** | **0.361** | **0.399** | **0.434** | **1.199** |

**Semi-supervised performances:** As $N$ decreases, an increase in MAE is observed across all models. This trend is particularly pronounced for the supervised models UMM-CSGM and MSDM†, which do not utilize samples with missing views. In contrast, it is worth noting that MDD performances only slightly decrease for $N \in \{70, 40, 10\}$. The semi-supervised adaptation MSDM†[N] shows similar results to its original formulation as $N$ decreases, indicating that the noisy formulation is not well-suited for producing highly faithful results. For UMM[N], the performance decrease is more substantial than for MDD, suggesting that the clean condition is not the sole factor contributing to MDD performances. At $N = 0\%$, all methods except MDD fail to learn the task. It is possible that sampling $t$ according to Eq. (4) may serve as an additional form of data augmentation.

**Bridge Translation:** The cube→(pyramid,icosphere) translation proves challenging, as can be seen in the qualitative images in Fig. 7, where the L1-map shows some shift of the object position, and in the Tab. 2 where the Bridge column has the highest error of all supervision setting, with an MAE of 1.199 compared to 0.316 for $N = 100\%$. Two factors may explain this: in the early stages of the backward diffusion process, the pyramid translation is suboptimal and subject to significant noise. In addition, the semantic inversion makes the generation more sensitive to noise, *e.g.* a wrong prediction in the pyramid color will also affect the prediction of the camera angle (see $\alpha$ in Tab. 1).

The intermediate bridge domain (pyramid) exhibits imperfect generation, with errors propagating to the subsequent icosphere translation. This challenge parallels the noisy condition issues discussed in Sec. 3.2, as the pyramid is unavailable for icosphere construction, similar to noisy translation models that lack access to the condition at the diffusion process initiation.

This challenge mirrors issues encountered in diffusion-based inpainting, where the condition is initially noisy (Chung et al., 2022; Lugmayr et al., 2022; Zhang et al., 2023a). Potential solutions from this field could be adapted to our context, such as resampling steps for domain synchronization (Lugmayr et al., 2022) or additional regularization costs (Zhang et al., 2023a; Chung et al., 2022), both approaches introduce additional computational overhead.

While this issue impacts the MAE on BL3NDT, the usual domain translation tasks do not expect a one-to-one mapping (*e.g.* face↔sketch translation), which mitigates the problem in real-world applications.

### 4.4 CelebAMask-HQ Results

To assess MDD's capability in generating realistic views for challenging domain-to-domain translation tasks, we evaluated domain translation between face, sketch, and segmentation mask on the CelebAMask-HQ dataset. This section considers the (sketch, mask)→face translation and shows that MDD can generate realistic faces.

The diversity of face generation, sketch generation, and segmentation generation is evaluated in Appendix B.8 along with additional quantitative and qualitative results.

As shown by Tab. 3, MDD closely matches LPIPS of UMM[N] in the different supervision settings and improves on SSIM. MDD significantly outperforms all other methods on both LPIPS and SSIM. In some

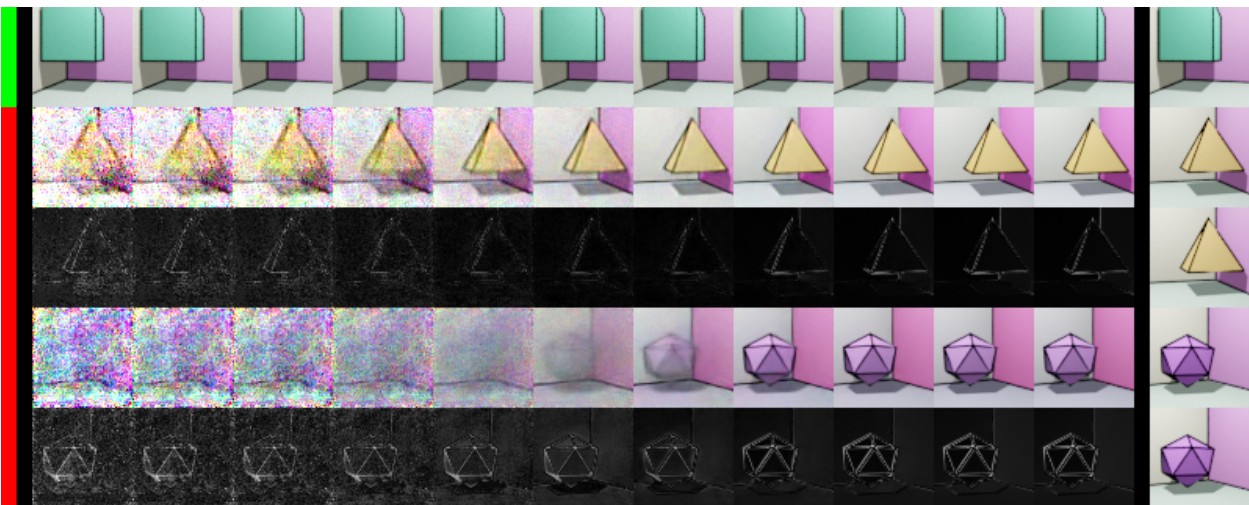

Figure 7: **Bridge translation on cube→(pyramid, icosphere).** The left row shows the condition in green and the generation in red. The right column shows the ground truth. Each row represents, from top to bottom, for each time in the backward diffusion process, the cube condition, the current pyramid translation, the current pyramid L1 map with the ground truth, the current iscophere translation, and the current icosphere L1 map with the ground truth.

instances, the UMM[N] method seems to produce oversaturated results but generally performs better than MSDM†[N]. This aligns with findings from other datasets, where the noisy condition paradigm consistently underperforms compared to the clean condition paradigm.

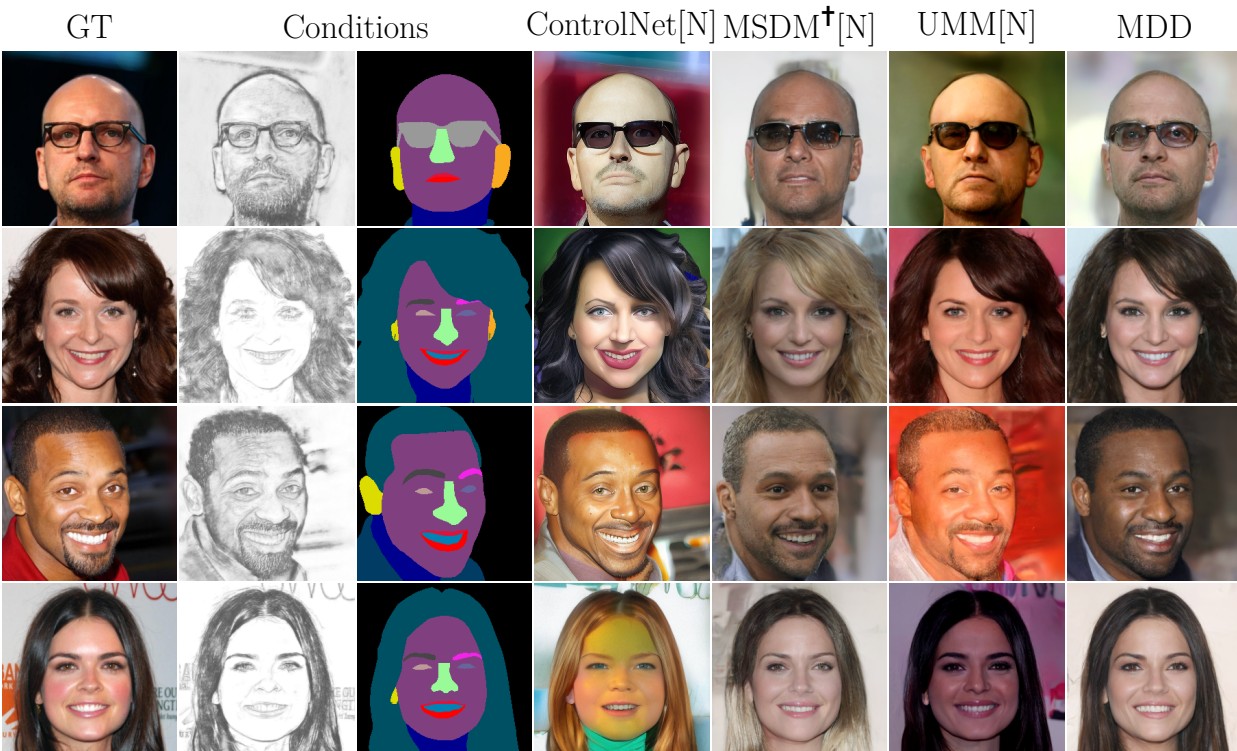

Figure 8: Generated faces given a sketch, mask condition, and the corresponding GT for N=0%.

Table 3: Evaluation metrics for (sketch, segmentation)→face on the CelebAMask-HQ dataset. 'ad.' refers to our adapted versions of state-of-the-art methods modified to handle semi-supervised scenarios.

| CelebAMask-HQ | | LPIPS↓ | | SSIM↑ | |
|---|---|---|---|---|---|
| | Face | 100% | 0% | 100% | 0% |
| ad. | MSDM†[N] | 0.3814 | 0.3785 | 0.1483 | 0.1306 |
| | UMM[N] | 0.2930 | 0.3500 | 0.2902 | 0.2177 |
| | MDD | **0.2305** | **0.3023** | **0.4385** | **0.3381** |

Table 4: Evaluation metrics for T1, T1ce, T2, and Flair, modalities completion on the BraTS 2020 dataset for different levels of supervision PN%, where N% indicates the probability of keeping a view. 'sota' refers to the original state-of-the-art methods without modifications, while 'ad.' refers to our adapted versions of these methods modified to handle semi-supervised scenarios.

| | | T1 | | | | | | T1ce | | | | | |
|---|---|---|---|---|---|---|---|---|---|---|---|---|---|
| | | PSNR↑ | | | SSIM↑ | | | PSNR↑ | | | SSIM↑ | | |
| | | P100% | P80% | P50% | P100% | P80% | P50% | P100% | P80% | P50% | P100% | P80% | P50% |
| sota | MSDM† | 21.623 | 18.583 | 11.077 | 0.883 | 0.735 | 0.146 | **27.159** | 25.971 | 19.067 | **0.911** | 0.840 | 0.135 |
| | UMM-CSGM | 22.806 | 17.769 | 13.280 | 0.894 | 0.680 | 0.064 | 26.947 | 26.488 | 26.558 | 0.910 | 0.898 | 0.892 |
| ad. | MSDM†[N] | 21.486 | 21.608 | 21.456 | 0.883 | 0.886 | 0.822 | 27.056 | 24.513 | 16.943 | 0.887 | 0.699 | 0.060 |
| | UMM[N] | 22.467 | 21.720 | 22.188 | 0.880 | 0.891 | 0.874 | 26.956 | 27.136 | 26.680 | 0.892 | 0.888 | 0.880 |
| | MDD | **23.004** | **23.373** | **22.928** | **0.902** | **0.903** | **0.892** | 27.052 | **27.567** | **27.285** | 0.894 | **0.909** | **0.892** |
| | | T2 | | | | | | Flair | | | | | |
| | | PSNR↑ | | | SSIM↑ | | | PSNR↑ | | | SSIM↑ | | |
| | | P100% | P80% | P50% | P100% | P80% | P50% | P100% | P80% | P50% | P100% | P80% | P50% |
| sota | MSDM† | 24.679 | 21.485 | 15.844 | 0.824 | 0.775 | 0.090 | 22.801 | 18.829 | 17.764 | 0.872 | 0.785 | 0.229 |
| | UMM-CSGM | 24.706 | 24.241 | 23.980 | 0.813 | 0.874 | 0.805 | 22.809 | 22.603 | 22.059 | 0.871 | 0.860 | 0.833 |
| ad. | MSDM†[N] | 25.249 | 20.895 | 12.321 | **0.891** | 0.701 | 0.029 | 23.491 | 21.649 | 12.366 | 0.839 | 0.753 | 0.042 |
| | UMM[N] | 24.726 | 24.843 | 24.036 | 0.882 | 0.887 | 0.861 | 23.110 | 23.384 | 22.726 | 0.845 | 0.840 | 0.827 |
| | MDD | **25.383** | **25.716** | **24.788** | 0.875 | **0.894** | **0.877** | **24.173** | **24.103** | **23.663** | **0.876** | **0.879** | **0.868** |

## 4.5 BraTS 2020 Results

### 4.5.1 Missing Modalities Completion

To demonstrate MDD's flexibility, we evaluate its performance in regenerating each modality on the BraTS 2020 dataset, establishing that a single trained MDD model successfully handles the generation of all modalities.

Table 4 presents quantitative results for PSNR and SSIM metrics, while Fig. 9 provides qualitative visual results. Additional quantitative analysis using other metrics (MSE, MAE) is provided in Appendix B.7. We focus primarily on T1 scan generation analysis and observe that our findings generalize to other modalities (Tab. 4) using only one model, highlighting the flexibility of multi-domain diffusion models compared to fixed diffusion models, which would require at least one model for each domain.

The existing formulations, MSDM† and UMM-CSGM, demonstrate significant performance degradation under reduced supervision conditions. Specifically, UMM-CSGM's PSNR on T1 generation decreases substantially from 22.81 to 13.28 when supervision is reduced from P100% to P50%, highlighting the inherent difficulty of this task with limited training data (Table 4). As illustrated in Fig. 9, these methods fail to generate clinically acceptable images outside the fully supervised setting. Our adaptation of these formulations (MSDM†[N] and UMM[N]) improves their resilience to supervision reduction, with both methods maintaining more stable performance metrics when supervision decreases, as evidenced in Tab. 4. Notably, MDD consistently outperforms the adapted UMM[N], despite the latter being specifically designed for missing scan completion tasks.

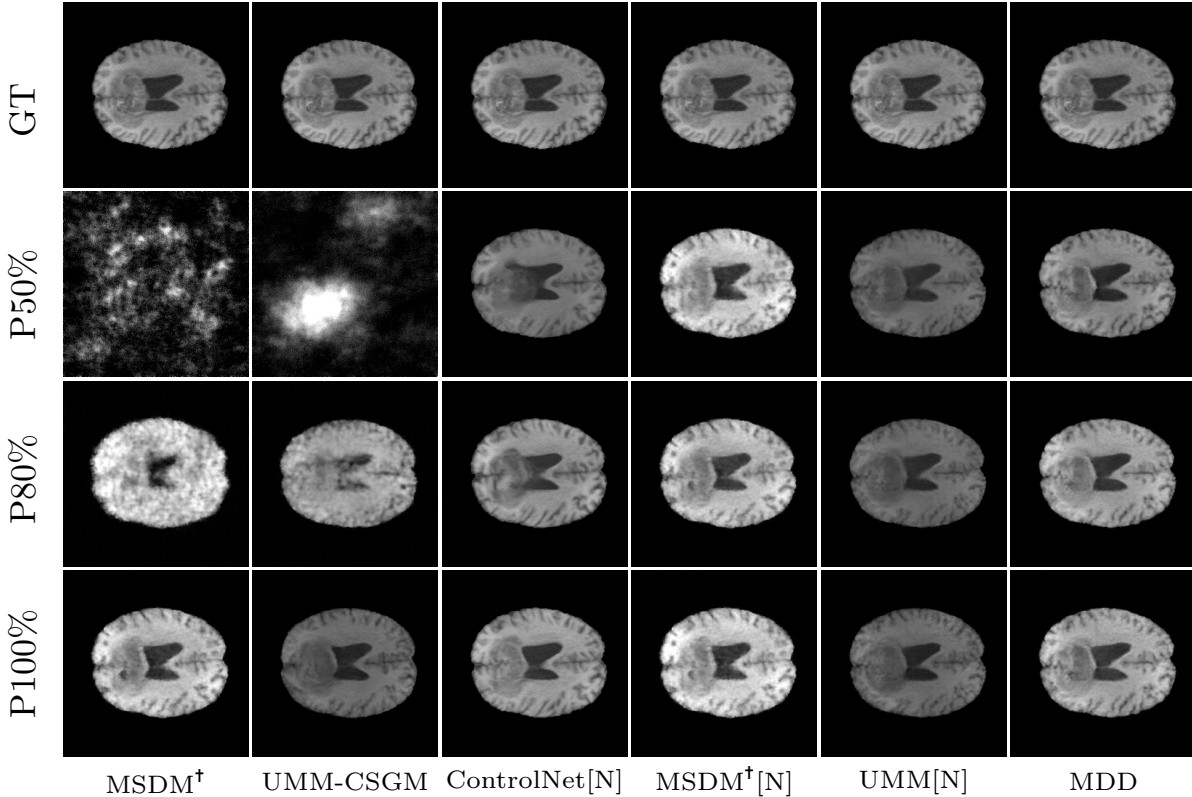

Figure 9: Generated T1 scan given all remaining modalities and the corresponding GT for different supervision levels.

### 4.5.2 Flexibility in the Number of Inputs

MDD demonstrates flexibility in handling both missing modality completion and varying numbers of condition domains during inference. We explore this adaptability through two experiments: missing modalities completion (focusing on T1 scan generation results here) and semantic segmentation (Appendix B.7), each with a gradually decreasing number of input modalities.

We gradually remove modalities in the order [Segmentation, T1ce, T2, Flair] and regenerate the missing domains, focusing on T1 scan generation. Qualitative results are presented in Fig. 10 and quantitative results in Tab. 5.

Models with noisy conditions demonstrate limitations in this setup. MSDM$^{\dagger}$[N]'s PSNR decreases by approximately 13% when the number of domains used drops from 4 to 1, compared to only a 4% decrease for MDD (Tab. 5). Generated T1 scans from MSDM$^{\dagger}$[N] are inconsistent when varying the number of condition domains, as observed in Fig. 10.

Conversely, models with clean conditions perform well. MDD not only outperforms UMM[N] but also maintains consistency as the number of available scans decreases. While UMM[N]'s PSNR shows a slight drop between four scans and one scan, its generation can be less consistent than MDD when the number of condition domains becomes very small.

The superior performance of MDD demonstrates how using different $t$ per domain allows for learning better data fusion compared to using a binary code (UMM[N]) or no code at all (MSDM$^{\dagger}$[N]). MDD effectively learns the relative information in different domains at each step, extracting data from less noisy domains to reconstruct the more noisy ones. These results indicate that MDD adapts well to varying numbers of inputs, particularly when condition domains become scarce. This suggests that MDD could be effectively used for

multimodal data fusion in situations where modalities may be missing during inference, offering a robust solution for diverse medical imaging scenarios.

Table 5: Evaluation metric for T1 scan generation on the BraTS 2020 dataset while varying the number of input domains. 'ad.' refers to our adapted versions of state-of-the-art methods modified to handle semi-supervised scenarios.

| BraTS 2020 | | PSNR↑ | | | | SSIM↑ | | | |
|---|---|---|---|---|---|---|---|---|---|
| T1 P50% | | 4 dom | 3 dom | 2 dom | 1 dom | 4 dom | 3 dom | 2 dom | 1 dom |
| ad. | MSDM†[N] | 21.456 | 21.336 | 20.615 | 18.711 | 0.822 | 0.817 | 0.806 | 0.752 |
| | UMM[N] | 22.188 | 22.175 | 22.055 | 21.758 | 0.874 | 0.878 | 0.870 | 0.852 |
| | MDD | **22.928** | **22.934** | **22.503** | **22.075** | **0.892** | **0.896** | **0.887** | **0.864** |

## 5 Limitations and Future Directions

In this section, we discuss some of the potential areas for improvement of MDD.

While MDD effectively handles the fusion of information from multiple domains (Sec. 4.5.2), the scalability of combining information from an increasing number of domains within a single latent vector could become a bottleneck. As the number of domains grows, the efficient integration and representation of multi-domain information may present computational and architectural challenges. The current architecture could benefit from more advanced data fusion strategies, particularly those designed for semi-supervised and imbalanced settings involving numerous domains (Han et al., 2024a). Such advanced fusion mechanisms could enhance the model's ability to handle heterogeneous data distributions and varying levels of annotation across domains.

Future work could explore the factorization of encoders and decoders across different modalities to improve scalability. Specifically, leveraging a powerful, general-purpose pretrained encoder could enable the extraction of domain representations without domain-specific training. This unified encoder could be conditioned on the specific domain it is processing through mechanisms similar to positional embeddings used in transformer architectures (Vaswani et al., 2017). Similarly, a single decoder could be employed to generate outputs across all domains using the processed latent code from domain fusion, given a general-purpose pretrained architecture. However, unlike the encoder, the decoder must be explicitly conditioned on the target domain since its input (the latent code) is domain-agnostic. This dual approach of shared encoder-decoder architecture with domain conditioning would facilitate scaling to multiple domains without a corresponding linear increase in model parameters at the encoder and decoder levels. While this approach might be well-suited for domain translation tasks where each domain can be represented in the same representation space (*e.g.* all domains of BL3NDT, BraTS 2020, and CelebAMask-HQ can be represented as images), it would present significant challenges for more heterogeneous domains such as image and text. In such cases, where the modalities differ fundamentally in their structure and dimensionality, this approach would not be directly applicable and would remain an open research challenge.

When working with fundamentally different modalities, MDD requires different encoders and decoders for each domain type, inherently limiting its scalability to a very large number of domains. A promising direction lies in the strategic sharing of parameters across different components of the model, leveraging the inherent similarities between different domains. This could be achieved through parameter sharing techniques, particularly through mixture-of-experts architectures (Wu et al., 2024; Han et al., 2024b; Li et al., 2025). Such approaches have the potential to efficiently scale to multiple domains while maintaining parameter efficiency through strategic sharing of weights across different domains. These methods could significantly reduce the computational overhead typically associated with multi-domain processing while preserving domain-specific expertise.

Another limitation of this study lies in the evaluation methodology. Although we employed diverse metrics that are widely accepted in the field, the incorporation of human preference evaluations could have provided

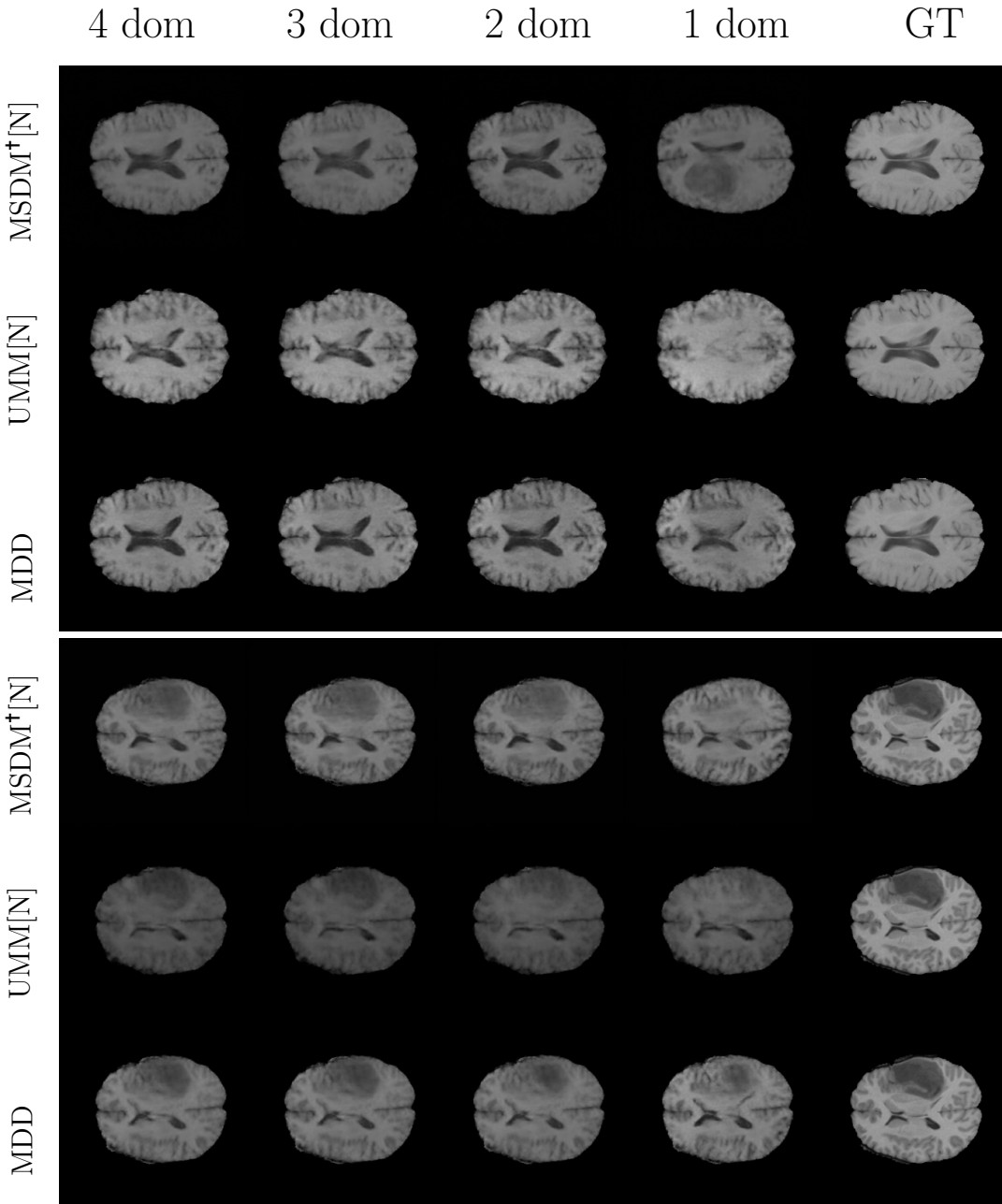

Figure 10: Generated T1 scans given multiple numbers of scans as input for P50%. Zoom in for better details.

additional valuable insights. However, such studies can be challenging to implement due to the need for humain annotators and standardized evaluation protocols.

## 6 Broader Impact Statement

As MDD is a generative framework requiring training datasets, it raises several important ethical considerations. These include concerns regarding image privacy, the inclusiveness of generated images which may exhibit biases, and the potential for memorization and reproduction of training examples.

MDD leverages synthetic datasets (Sec. 4.1.1) containing geometric forms, which inherently circumvents issues related to data privacy, biases, and inclusiveness. The additional datasets employed in this research are publicly available, thereby mitigating certain data rights and privacy concerns. Nevertheless, several critical aspects require careful attention, including potential biases in image generation, demographic representation, and training data memorization and reproduction.

To address these challenges, we recommend the following measures: First, it is crucial to ensure proper data usage rights and permissions for all training images. Second, the training dataset should be carefully curated to represent diverse populations and scenarios, avoiding demographic or contextual biases. Third, thorough analysis of the generated image distribution should be conducted to detect and address any learned biases or potential privacy breaches. Additionally, regular auditing of the model's outputs and systematic evaluation of its societal impact should be performed to ensure responsible deployment of the technology.

## 7 Conclusions

We demonstrate that the MDD framework effectively learns multi-modal, multi-domain translation in a semi-supervised setting by modeling domain-specific noise levels. This approach facilitates domain translation learning without requiring the definition of a specific condition domain.

Unlike existing frameworks, MDD can handle a significant number of modalities without substantially increasing model size. Moreover, it exhibits superior information fusion from different modalities, making it particularly suitable for tasks with a large number of domains.

Our research elucidates how MDD unifies various approaches to applying noise to the condition in domain translation tasks. We found that maintaining a clean condition during generation yields excellent results, as it allows leveraging the condition from the start of the diffusion process. Using a noisy condition is particularly detrimental when tasks involve deterministic mapping, especially when the target domain is a semantic segmentation map. We attribute this issue to the diffusion model's need to generate a mean image without condition information at the start of the generation process, and subsequently correct erroneous predictions.

Our work integrates seamlessly with existing literature on diffusion frameworks that aim to learn multi-modal domain translation without defining a specific translation path (Meng et al., 2022b; Bao et al., 2023; Mariani et al., 2024) and by applying different noise levels per modality (Meng et al., 2022b; Mariani et al., 2024). Extending these existing works, we demonstrate that the proposed formulation allows for semi-supervised conditional domain translation. This can reduce the data burden in settings where data is difficult to acquire, such as the medical field, and allow for flexible translation with different inputs.

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

# Supplementary Material

## A  Implementation Details

### A.1  Diffusion Model

During training, we use a linear noise schedule or $(1e-4, 2e-2)$ and the noise prediction function described in Sec. 3.1. The maximum number of diffusion steps $T$ is $1\,000$. During generation, DDIM (Song et al., 2021) with 100 steps is used for BL3NDT, DDPM (Ho et al., 2020) with 1000 steps for BraTS 2020, and DDPM with 1000 steps for CelebAMask-HQ when faces are part of the generation; otherwise, DDIM with 250 steps for the other CelebAMask-HQ modalities.

### A.2  Training Details

#### A.2.1  Multi-Domain Translation Models

For all multi-domain translation models (MSDM†, UMM-CSGM, and MDD), we employ consistent hyperparameter settings within each dataset. We utilize the Adam optimizer with an initial learning rate of $7 \times 10^{-5}$ for BraTS 2020 and $2 \times 10^{-5}$ for both BL3NDT and CelebAMask-HQ datasets. The Adam parameters $\beta_1$ and $\beta_2$ are set to 0.9 and 0.999, respectively. For BL3NDT and BraTS 2020, we implement a learning rate decay strategy, multiplying the rate by 0.75 every 10 epochs. We employ batch sizes of 128 for BL3NDT, and 64 for both BraTS 2020 and CelebAMask-HQ. All models are trained with Exponential Moving Average (EMA) on model parameters with a decay rate of 0.9999. For BL3NDT, we implement early stopping with a patience of 40 epochs and a maximum computational budget of 240 epochs. For BraTS 2020, all models are trained for 200 epochs, while for CelebAMask-HQ, training continues for 2000 epochs.

#### A.2.2  Uni-Direction Translation Models

For ControlNet implementations (Zhang et al., 2023b), we use the publicly available code repository and select each model based on its best validation metrics. For the BL3NDT dataset, we train with an unlocked decoder for 1000 epochs with a batch size of 128. For BraTS 2020, training proceeds with an unlocked decoder for 700 epochs using a batch size of 128. For CelebAMask-HQ, we conduct training for 2000 epochs with a batch size of 1024.

### A.3  Data Normalization

For BL3NDT, images are linearly normalized between [-1,1], and no data augmentation is applied.

For BraTS 2020, each 3D scan is independently linearly normalized between [0,1], and the semantic segmentation classes are merged to create a binary segmentation and are one hot encoded with one channel. The bottom 80 and top 26 slices are removed. Each resulting 3D volume is sliced in the axial axis and resized to (224, 224).

For CelebAMask-HQ, faces and sketches are linearly normalized in [-1,1], and the 19 classes are one hot encoded. We resize each modality in (256,256) and use random horizontal flips as data augmentation.

### A.4  Model Architecture and Adaptation to Multiple Domains

All models use the same U-Net architecture based on (Ho et al., 2020) with some modifications to accommodate multiple domains and times, illustrated in Fig. 11. We call $E$ the U-Net encoder, $B$ its bottleneck, and $D$ its decoder. In a setting with $L$ domains, the encoder and decoder are duplicated $L$-times and function as in the classical single-domain diffusion setting, *i.e.* the encoder $E^{(l)}$ takes as input the modality $x^{(l)}$ and the time $\mathcal{T}^{(l)}$ and produces its embedding $e^{(l)}$ and a list of skip connections skips$^{(l)}$. The list of embeddings $[e^{(1)}, ..., e^{(L)}]$ and times $\mathcal{T}$ is then processed by a bottleneck adapted for multi-domain multi-time, which we will describe later, to produce the processed embedding $emb$. Each decoder $D^{(l)}$ takes as input the processed

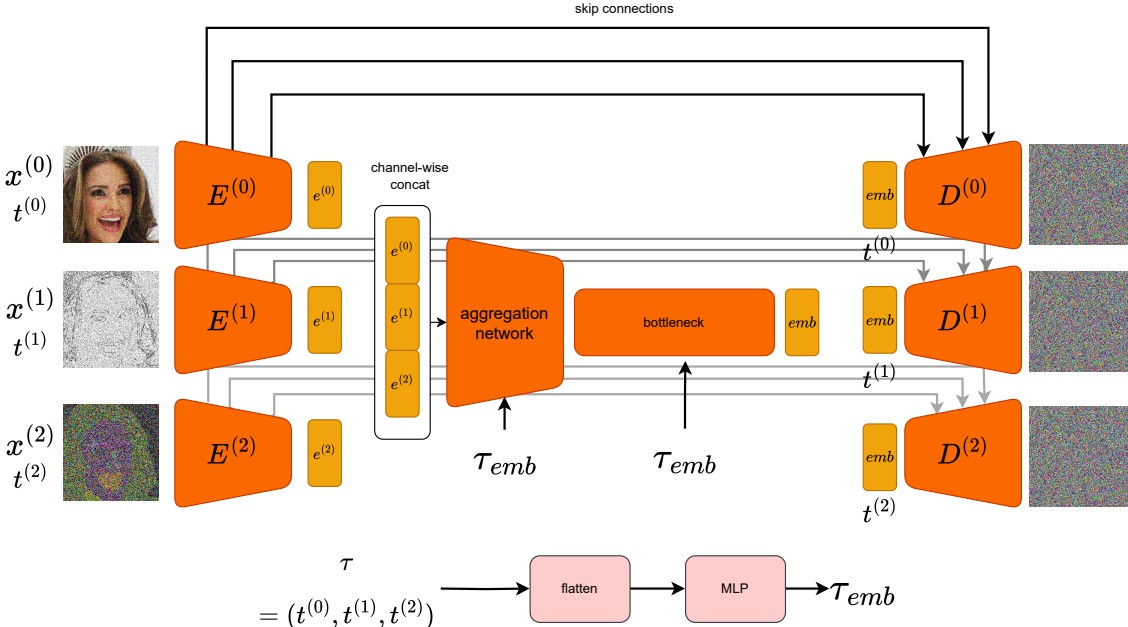

Figure 11: Illustration of the MDD architecture with a UNet backbone on 3 domains. Each domain has its own encoder $E^{(i)}$ and decoder $D^{(i)}$ that processes modalities individually using their associated timesteps $t^{(i)}$. The embeddings from each modality are concatenated and aggregated for processing through the bottleneck. Subsequently, the processed embeddings are passed through individual decoders $D^{(i)}$, with skip connections from their corresponding encoders $E^{(i)}$ to predict the noise $\epsilon^{(i)}$ added to each domain. In the aggregation network and bottleneck, time embeddings are first flattened, then processed through an MLP for scale-shift operations, following standard practices.

embedding $emb$, the list of skips connections $\text{skips}^{(l)}$ and time $\mathcal{T}^{(l)}$, and predicts the noise map for the modality $x^{(l)}$ as in the classical single-domain diffusion setting.

To accommodate multiple domains, an aggregation network is added before the bottleneck, taking as input the list of embeddings $[e^{(1)}, ..., e^{(L)}]$ each of shape $(b, c, h, w)$ and times $\mathcal{T}$. The embeddings are concatenated on the channel dimension before going through a ConvNextBlock which reduces their dimension from $c \times L$ back to $c$. They are then processed through the bottleneck layers as the original U-Net would.

Each layer that uses the entire $\mathcal{T}$ is modified. The $\mathcal{T}$ of shape $(b, L)$ is embedded once at the beginning of the forward pass using the initial time-embedding MLP into a tensor of shape $(b, L, tdim)$, then reshaped into shape $(b, L \times tdim)$. Then, each bottleneck ConvNextBlock time MLP input dimension is modified to take as input a vector of shape $(b, L \times tdim)$ instead of taking a vector of shape $(b, tdim)$.

## B    Additional Experiments

### B.1    Exploring the Noise on Condition Strategy

During MDD training, there is no domain identified as a condition domain or as a target domain, so the question is how to use the condition during the generation process: should the condition be kept clean (using $x_{\text{cond},0}$, see Algorithm 2) or should it contain the same level of noise as in generation (using $x_{\text{cond},t}$, see Algorithm 2). We experiment on BL3NDT dataset to compare the effect of different noise levels during the generation process and define four noise strategies for the condition described in Fig. 12. We define $\phi_\gamma(t)$ as the function that allows to obtain the noise level in the condition domains according to the noise level present in the generated domains.

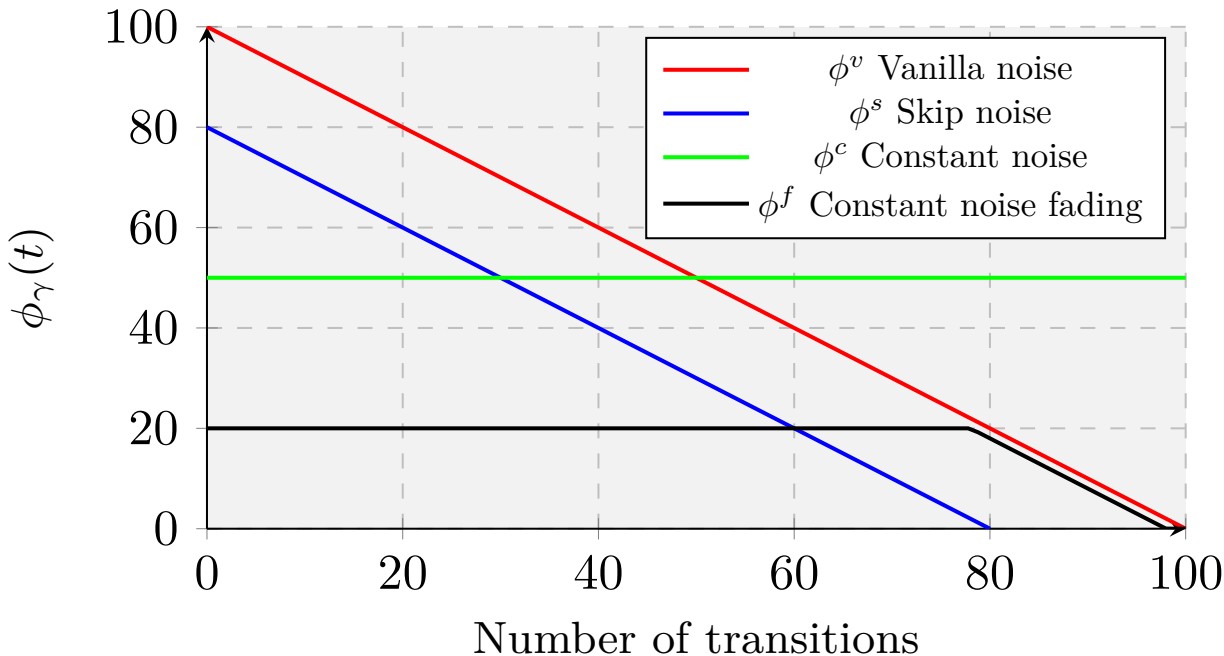

Figure 12: **Illustration of different $\phi_\gamma(t)$ functions.** Evolution of the noise applied to the condition as a function of the number of transitions made for the target domain. Vanilla diffusion steps have the drawback of not properly preserving semantic information of the condition and requiring multiple jumps. Skip Noise diffusion removes information of the condition in the same way as the Vanilla diffusion step, but allows a cleaner condition at the start, therefore better preserving semantic information from the condition. Constant noise continuously removes the same amount of information at each step. We found this solution to work best in practice with a low amount of 20% of the total number of time steps. We also tried applying constant noise to the condition until the noise level of the target domain caught up with the condition.

Approaches that learn the joint distribution apply the same noise level on each modality during training (Mariani et al., 2024) and thus also during generation. This strategy is called Vanilla noise because it closely follows the original backward diffusion process. When a condition domain is identified, it is often kept clean during training and generation (Xie et al., 2024; Cross-Zamirski et al., 2023; Saharia et al., 2022a; Lyu & Wang, 2022). We identify this strategy as Constant Noise, which can be parameterized by a noise level, a clean condition is identified as Constant Noise(0). We also explore two additional strategies: Skip Noise, which is vanilla noise, where the noise applied to the condition is less than that applied to the generation, and Constant Noise Fading, where the noise remains constant until it catches up with the generation.

For the BL3NDT dataset with the Bridge data setting, we found that the higher the $\gamma$, the lower the MAE. This is not surprising, as the higher the $\gamma$, the less noisy the condition, allowing the model to use information from the condition early in the generation process and avoid drifting too far from the correct semantic. Interestingly, the $\phi$ Skip Noise function has a low MAE even for a relatively high level of $\gamma$. We speculate that this noise scheme still allows the diffusion model to focus first on the low-level frequency that is erased from the condition and, later on, the high-level frequency details.

### B.2 Ablation Study

To extensively test MDD capabilities, further ablation is provided by removing the contribution of Eq. (4), MDD training scheme is adapted to noisy conditions (named MDD[NOISY]). $\mathcal{T}$ is set as $\mathcal{T} = [t, t, t]$ during training. During generation, missing domains are replaced with noise and their $t$ with $T$ as specified in Eq. (6). Ablation of the additional step in Algorithm 1, line 9 is also provided (named MDD[RAND]). The different results tables refer to these adapted models as 'ours'.

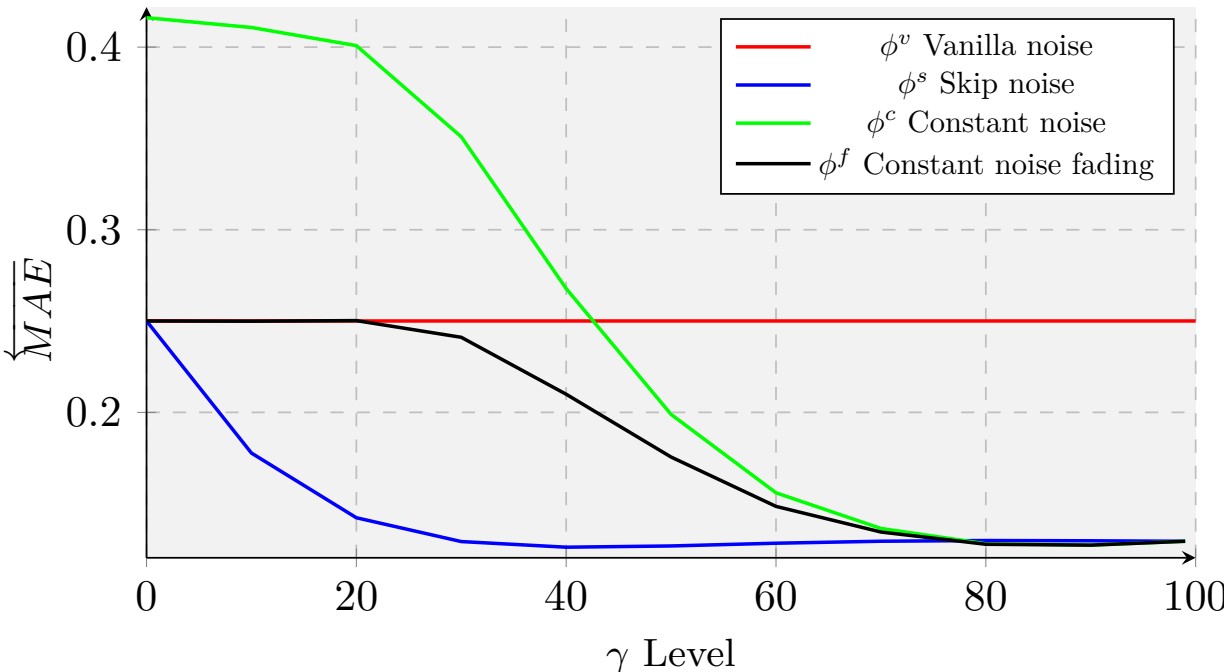

Figure 13: Comparison of the MAE↓ for different $\phi_\gamma(t)$ functions according to the $\gamma$ parameter on BL3NDT dataset using the Bridge data setting. In general, the smaller the $\gamma$, the higher the MAE. The smaller $\gamma$, the noisier the condition, and the harder it is for the model to use the information within the condition in the early steps.

**Noisy generation: MDD[NOISY].** Using a noisy condition instead of a clean condition significantly increases the translation error, as shown in Tab. 6. This is more true for the BL3NDT dataset than for the other datasets, because the BL3NDT dataset expects a specific output with challenging variable inversions between domains. For other datasets, such as BraTS 2020, MDD also consistently outperforms MDD[NOISY], as shown in Tabs. 13 to 16.

For translation tasks where the target is semantic segmentation, MDD[NOISY] fails the task as shown in Tabs. 17 and 19 by producing realistic segmentation but unaligned with the condition, which is consistent with other noisy condition models such as MSDM†[N] (Fig. 22). When generating segmentation maps from other condition domains, MDD[NOISY] (and other noisy condition models) has no mechanism to distinguish which domain is part of the generation and which is part of the condition; therefore, correcting the target domain (in this case, the segmentation) becomes particularly challenging. A similar problem has been reported in other works (Lugmayr et al., 2022; Chung et al., 2022) where the current generation and condition are not well synchronized. Proposed solutions involve resampling mechanisms (Lugmayr et al., 2022; Chung et al., 2022), which increases computation time, making prediction of many images impractical for most institutions.

This validates the assumption that using a clean condition allows diffusion models to leverage the condition from the early steps of diffusion, leading to more faithful and realistic generations.

**Condition vector $m_c$: MDD[RAND].** Tweaking the noise scheduling during training allows MDD to better follow a specific set of domains during generation. We found that this formulation is beneficial for tasks where a specific output is expected, such as the BL3NDT dataset (Tab. 6) and segmentation tasks (Tabs. 17 and 19). For other translation tasks, MDD performs better than MDD[RAND], but with a smaller metric difference (Tabs. 13 to 16).

Table 6: BL3NDT ablation, MAE↓ error (values are given in e-1 order) for different amounts of supervision on the BL3NDT dataset for cube→(pyramid, icosphere) translation.

| | BL3NDT MAE↓ | 100% | 70% | 40% | 10% | 0% | Bridge |
|---|---|---|---|---|---|---|---|
| | MDD[NOISY] | 1.977 | 2.001 | 2.030 | 2.158 | 2.740 | 4.022 |
| ours | MDD[RAND] | 0.616 | 0.652 | 0.715 | 0.803 | 0.565 | 1.294 |
| | MDD | **0.316** | **0.338** | **0.361** | **0.399** | **0.434** | **1.199** |

## B.3 Sampling Strategy Impact on Diversity

To further validate the flexibility of MDD in modeling different noise levels, we evaluate the noise strategies presented in Appendix B.1 applied to CelebAMask-HQ for face translation with sketch and segmentation conditions using full supervision. Results are presented in Fig. 14. For computational efficiency, we employ DDIM with 100 steps instead of DDPM with 1000 steps. While this results in a slight degradation of LPIPS compared to the results presented in Tab. 3, the performance remains superior to other baselines.

For most $\phi_\gamma$ strategies, a modest increase in $\gamma$ leads to decreased image quality (LPIPS) and, notably, without an increase in image diversity. This phenomenon may be attributed to the initial noise injection removing important features from the sketch that affect LPIPS computation, while being insufficient to introduce meaningful diversity in the generation process. Increasing noise injection into the condition enables greater image diversity, albeit at the cost of image quality, establishing a clear trade-off between these metrics. Fundamentally, these metrics are inherently antagonistic in domain translation tasks: while we aim to generate samples that closely match the ground truth, perfect fidelity would result in zero diversity, making it challenging to optimize both metrics simultaneously.

Further research is needed to develop additional mechanisms for MDD to achieve better control over generation diversity. One potential approach would be to implement a progressive condition relaxation during generation, whereby the condition transitions from a fixed constraint to becoming part of the generation domain. This controlled transition would enable the generated output to gradually deviate from the initial condition, offering a more structured approach to diversity than simple noise injection.

## B.4 Sensitivity to Noise Schedulers

We evaluate MDD's adaptability to different noise schedulers by examining its performance across multiple scheduling strategies. In our experiments, we employ DDPM (Ho et al., 2020) for BraTS 2020 and CelebAMask-HQ datasets, while utilizing DDIM (Song et al., 2021) for BL3NDT to optimize computational efficiency. Both DDPM and DDIM implement linear noise schedules. We further extend our analysis by retraining the model with the cosine scheduler (Nichol & Dhariwal, 2021), which provides more gradual noise application at higher noise levels. Results are presented in Tab. 7 for the Bridge data setting on BL3NDT and in Tab. 8 for the P50% data setting on BraTS 2020, focusing on T1 scan generation.

MDD demonstrates robust performance across different noise schedulers, with the cosine scheduler achieving comparable MAE scores to DDIM with a linear scheduler on BL3NDT, and even surpassing previous results with fewer sampling steps. Similar behavior is observed for the BraTS 2020 dataset, where the cosine scheduler performs as well as the linear scheduler while requiring only a fraction of the sampling steps.

These findings suggest that MDD is robust to different scheduler choices and could potentially leverage recent advances in more sophisticated scheduling strategies.

Notably, significantly reducing the number of diffusion steps does not substantially degrade generation performance. For BL3NDT, and particularly for BraTS 2020, reducing the steps from 1000 to 50 produces minimal differences in SSIM scores. This resilience might be attributed to the mitigation of exposure bias (Ning et al., 2023) at inference time, where the distribution of inputs typically deviates from the training distribution and errors accumulate during diffusion steps. Fewer diffusion steps in DDIM may help alleviate this issue by reducing the cumulative error, while the noise reinjection process in DDPM provides additional regularization during sampling.

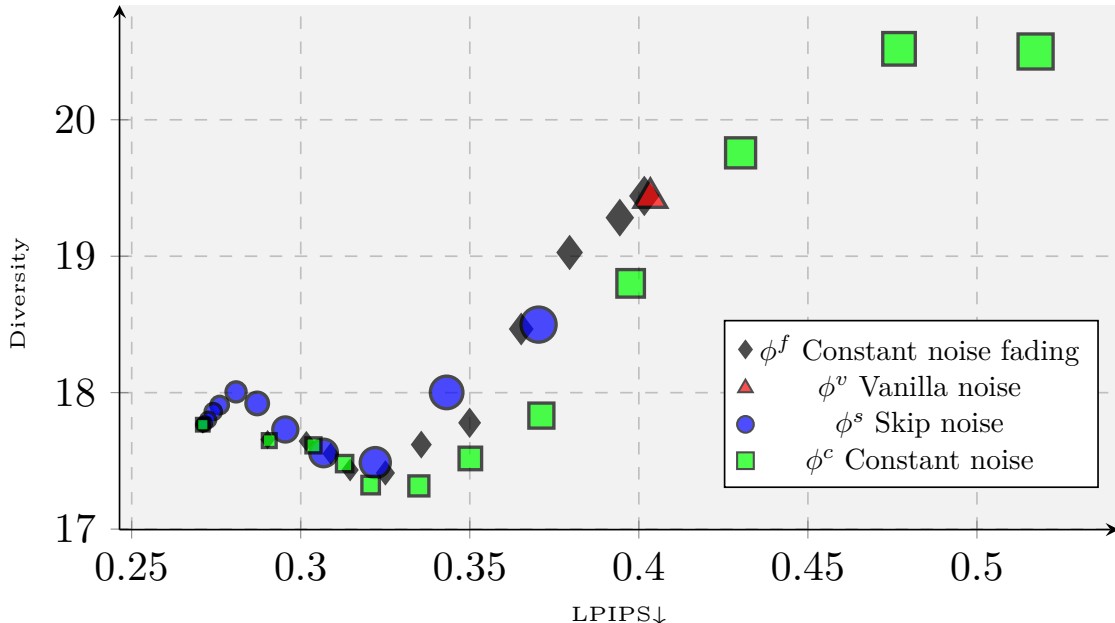

Figure 14: Trade-off between diversity and LPIPS across different noise sampling strategies $\phi_\gamma$ while varying the noise level parameter $\gamma$. Marker sizes correspond to $\gamma$ values, where larger markers indicate lower $\gamma$ values, representing increased noise injection into the condition. Note that $\gamma$ values cannot be directly compared across different noise sampling strategies $\phi_\gamma$, as each strategy implements this parameter distinctly.

Table 7: MDD performance with different noise schedulers on BL3NDT Bridge setting for cube→(pyramid, icosphere) translation.

| Sampler | Number Of Steps | MAE↓ |
|---|---|---|
| DDIM Cosine | 50 | 0.974 |
| DDIM Cosine | 100 | 0.979 |
| DDIM Linear | 100 | 1.199 |

## B.5 MDD Computational Analysis

We analyze MDD's computational characteristics in terms of generation speed, parameter count, floating-point operations (FLOPs), and multiply-accumulate operations (MACs). FLOPs and MACs are measured using Calflops[3] for a single complete forward pass of the model. Generation time is computed over 50 diffusion steps for 1 image and averaged across 100 runs on NVIDIA A100 GPU. We denote the time required for translating between $n$ domains as $n$DT in seconds. The translation time for $n$ domains using ControlNet is computed by multiplying the time required for a single domain generation by $n$, as domains must be generated sequentially rather than simultaneously. Results are presented in Tab. 9.

The multi-domain translation baselines (MDD, MSDM†, UMM-CSGM) demonstrate comparable generation speeds, as they share a similar UNet backbone architecture and generate all missing domains simultaneously. Consequently, the generation time for $n$ domains remains constant across these models. In contrast, the single-domain translation baseline ControlNet, although faster for individual domain generation, may become less efficient in scenarios requiring multiple output domains, as it necessitates separate generation processes for each domain.

For $n = 1$, ControlNet achieves a generation time of 7.5s, approximately 2.7 times faster than MDD (20.1s). This difference can be attributed to ControlNet operating in the latent space, while MDD operates in the

---

[3] https://github.com/MrYxJ/calculate-flops.pytorch

Table 8: MDD performance with different noise schedulers on BraTS 2020 P50% setting for T1 scan generation.

| Sampler | Number Of Steps | PSNR↑ | SSIM↑ |
|---------|-----------------|-------|-------|
| DDIM Cosine | 50 | 21.699 | 0.894 |
| DDIM Cosine | 100 | 22.387 | 0.881 |
| DDPM Linear | 1000 | 22.928 | 0.892 |

pixel space. Operating in the latent space could potentially improve MDD's performance, and we plan to extend our approach to incorporate latent space operations in future work.

For $n = 2$, MDD (20.1s) is slightly slower than ControlNet (14.9s), and for $n \geq 3$, where MDD is designed to operate, it outperforms ControlNet in terms of generation time. The efficiency gap between MDD and uni-domain generation approaches like ControlNet widens as the number of domains increases. For instance, with $n = 5$, MDD (20.1s) is 1.9 times faster than ControlNet (37.3s).

Additionally, ControlNet's multi-domain generation ($n \geq 2$) relies on using generated domains as conditions for subsequent generations, which may introduce domain adaptation issues and error accumulation across multiple generations, potentially limiting its practicality in such scenarios.

Furthermore, though not quantified in our analysis, the storage and training time requirements for ControlNet should be considered, as each translation configuration requires a separate model to be trained and maintained.

Table 9: Computational complexity comparison across different domain translation methods. We report parameter count, FLOPs, MACs, and generation time ($n$DT) in seconds for translating $n$ domains. All measurements are performed on NVIDIA A100 GPU.

| Model | TFLOPs | MACs | Params | 1DT | 2DT | Multi-Domain Translation 3DT | 4DT | 5DT |
|-------|--------|------|--------|-----|-----|-----|-----|-----|
| MDD | 1.4795 | 733.556 GMACs | 266.639 M | 20.1 | 20.1 | 20.1 | 20.1 | 20.1 |
| UMM-CSGM | 1.4795 | 733.556 GMACs | 266.639 M | 20.6 | 20.6 | 20.6 | 20.6 | 20.6 |
| MSDM† | 1.4795 | 733.556 GMACs | 266.639 M | 20.1 | 20.1 | 20.1 | 20.1 | 20.1 |
| ControlNet | 24.8474 | 12.4135 TMACs | 1.2301 B | 7.5 | 14.9 | 22.4 | 29.8 | 37.3 |

## B.6 Comparison to Model with a Fixed Configuration

We compare MDD with ControlNet (Zhang et al., 2023b), a diffusion-based domain translation framework that enables translation from a fixed set of conditional domains to a specific target domain. While **ControlNet is limited to one specific configuration among the $2^n$ possible translation configurations**, making it inherently different from our framework's flexibility, we include this comparison to demonstrate the advantages of our approach.

### B.6.1 Translation With Complex Semantic Inversion

The translation cube→(pyramid, icosphere) is evaluated. For this purpose, ControlNet requires training two separate models: one for cube→pyramid and another for cube→icosphere. In the bridge supervision setting (where 50% of (cube, pyramid) pairs and 50% of (pyramid, icosphere) pairs are available), two distinct models must be trained: cube→pyramid and pyramid→icosphere. Moreover, for the bridge translation scenario, using pyramid→icosphere requires generated pyramid images as conditions, which can introduce domain adaptation challenges due to potential distribution shifts between generated and real images. This demonstrates a significant limitation of ControlNet, as different models must be trained for different configurations, whereas MDD efficiently handles all configurations using a single unified model.

Table 10: Evaluation of translation domain with a fixed-configuration using ControlNet. MAE↓ error (values are given in e-1 order) for different amounts of supervision on the BL3NDT dataset for cube→(pyramid, icosphere) translation.

| BL3NDT MAE↓ | 100% | 70% | 40% | 10% | 0% | Bridge |
|---|---|---|---|---|---|---|
| ControlNet[N] | 1.707 | 1.946 | 2.116 | 2.315 | 2.408 | 2.390 |
| MDD | **0.316** | **0.338** | **0.361** | **0.399** | **0.434** | **1.199** |

Table 11: Evaluation of translation domain with a fixed-configuration using ControlNet. Evaluation metrics for T1 modality completion on the BraTS 2020 dataset for different levels of supervision PN%, where N% indicates the probability of keeping a view.

| BraTS 2020 | PSNR↑ | | | SSIM↑ | | |
| T1 | P100% | P80% | P50% | P100% | P80% | P50% |
|---|---|---|---|---|---|---|
| ControlNet[N] | **24.929** | **24.665** | **23.968** | 0.890 | 0.869 | 0.832 |
| MDD | 23.004 | 23.373 | 22.928 | **0.902** | **0.903** | **0.892** |

We report the MAE results in Tab. 10. Despite training with an unlocked stable diffusion decoder [4], ControlNet[N] fails to effectively solve the translation task. While it generates plausible and visually coherent images, it struggles significantly with semantic inversion and frequently misassigns colors (Fig. 15). Specifically, ControlNet[N] accurately positions the geometric objects but consistently applies incorrect colors to either the objects or the surrounding walls. Unlocking the full model (encoder, bottleneck, and decoder instead of just the decoder) resolves this issue but requires much more training time and computations; this reveals an inherent limitation in ControlNet's domain translation capabilities. These results suggest that, unlike MDD, ControlNet's capabilities are limited to tasks with strong pixel-to-pixel correspondences and may not be suitable for more complex domain translation tasks requiring semantic understanding.

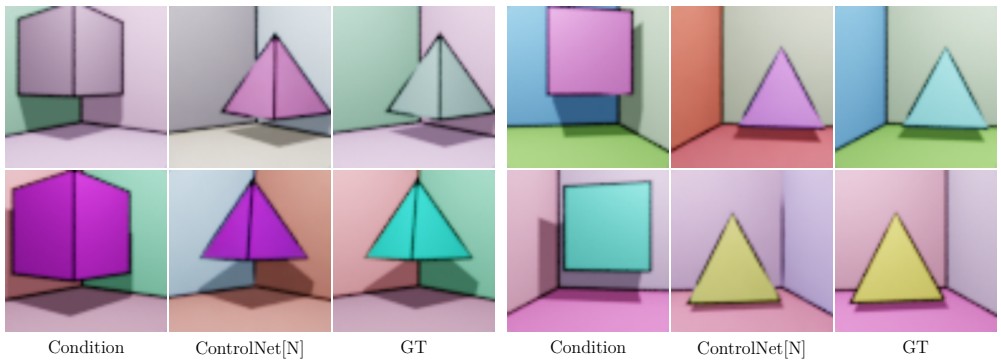

Figure 15: Evaluation of translation domain with a fixed-configuration using ControlNet. Translation on cube→pyramid for $N = 100$ on BL3NDT.

### B.6.2 Missing T1 Completion on BraTS 2020

For the BraTS 2020 dataset, ControlNet[N] demonstrates strong performance (Fig. 9) when adapted to the semi-supervised setting, achieving high PSNR values but lower SSIM scores compared to MDD (Tab. 11). Since SSIM evaluates the similarity in luminance, contrast, and structural information between images, it provides a more clinically relevant assessment for medical scan evaluation, where preservation of anatomical structures is critical for diagnostic purposes.

---

[4]https://github.com/lllyasviel/ControlNet/blob/main/docs/train.md#sd_locked

### B.6.3   Face Generation on CelebAMask-HQ

To complement our multi-domain translation evaluation (Sec. 4.4), we assess the (sketch, mask)→face translation by training one ControlNet[N] for each supervision proportion. For $N = 100\%$, ControlNet[N] generates realistic images as evidenced by low LPIPS and SSIM metrics (Tab. 12). It can be observed that MDD produces better metrics than ControlNet[N], despite ControlNet[N] leveraging powerful pre-trained model which, which show MDD capacity to fully utilize the cross-domain information during training. However, for $N = 0\%$, despite being trained in the same semi-supervised setting as MSDM$^{\dagger}$[N] and UMM[N] (with missing samples replaced by noise), ControlNet[N] fails to adapt effectively to the translation task, as indicated by significantly higher LPIPS values and poor visual quality (Fig. 8). This highlights how fixed-domain translation conditioning mechanisms demonstrate reduced robustness in challenging low-supervision settings (when no fully supervised training data exists, unlike for the BraTS 2020 setting) without specific adaptations. In contrast, our model produces highly realistic images even under these constrained conditions (Fig. 8), demonstrating superior generalization capabilities.

Table 12: Evaluation of translation domain with a fixed-configuration using ControlNet. Evaluation metrics for (sketch, segmentation)→face on the CelebAMask-HQ dataset.

| CelebAMask-HQ | LPIPS↓ | | SSIM↑ | |
|---|---|---|---|---|
| Face | 100% | 0% | 100% | 0% |
| ControlNet[N] | 0.2391 | 0.42780 | 0.4330 | 0.1833 |
| MDD | **0.2305** | **0.3023** | **0.4385** | **0.3381** |

### B.7   BraTS 2020 Additional Missing Modalities Completion Results

Additional quantitative and qualitative results are provided for all missing modalities on BraTS 2020 for a more detailed analysis of the missing modality completion task. Additional results for T1 generation Tab. 13 and Fig. 16, results for T1ce generation Tab. 14 and Fig. 17, T2 generation Tab. 15 and Fig. 18 and Flair generation Tab. 16 and Fig. 19 are consistent with those presented in the main paper, where MDD performs strongly on metrics considered.

For segmentation generation Tab. 17 and Fig. 20, we found that noisy generation strategies (MSDM$^{\dagger}$ and MSDM$^{\dagger}$[N]) are unable to perform the task even when a lot of supervision is available (Fig. 20). It is possible that at the beginning of the generation, when the condition is noisy, MSDM$^{\dagger}$ and MSDM$^{\dagger}$[N] are unable to predict a meaningful "mean" due to the binary and localized nature of the segmentation maps. However, this effect may be mitigated for scan modalities as it can predict a correct mean image that better represents the scan modality distribution.

Table 13: Evaluation metrics for T1 modality completion on the BraTS 2020 dataset. To save place, SSIM values are given in e+1 order, MAE values are given in e-2 order, and MSE values are given in e-3 order. 'sota' refers to the original state-of-the-art methods without modifications, while 'ad.' refers to our adapted versions of these methods modified to handle semi-supervised scenarios.

| | BraTS 2020 | PSNR↑ | | | SSIM↑ | | | MAE↓ | | | MSE↓ | | |
|---|---|---|---|---|---|---|---|---|---|---|---|---|---|
| | T1 | P100% | P80% | P50% | P100% | P80% | P50% | P100% | P80% | P50% | P100% | P80% | P50% |
| sota | MSDM$^{\dagger}$ | 21.62 | 18.58 | 11.08 | 8.83 | 7.35 | 1.46 | 3.31 | 5.19 | 17.46 | 6.88 | 13.86 | 78.03 |
| | UMM-CSGM | 21.49 | 21.61 | 21.46 | 8.83 | 8.86 | 8.22 | 3.34 | 3.28 | 3.54 | 7.10 | 6.91 | 7.15 |
| ad. | MSDM$^{\dagger}$[N] | 22.81 | 17.77 | 13.28 | 8.94 | 6.80 | 0.64 | 2.92 | 5.80 | 14.76 | 5.24 | 16.71 | 46.99 |
| | UMM[N] | 22.47 | 21.72 | 22.19 | 8.80 | 8.91 | 8.74 | 3.06 | 3.34 | 3.18 | 5.67 | 6.73 | 6.04 |
| ours | MDD[RAND] | **23.44** | 23.15 | 21.70 | 8.84 | 8.83 | 8.63 | **2.72** | 2.79 | 3.32 | **4.53** | 4.84 | 6.76 |
| | MDD[NOISY] | 21.85 | 21.99 | 21.61 | 8.85 | 8.80 | 8.78 | 3.19 | 3.18 | 3.30 | 6.52 | 6.32 | 6.91 |
| | MDD | 23.00 | **23.37** | **22.93** | **9.02** | **9.03** | **8.92** | 2.84 | **2.73** | **2.87** | 5.01 | **4.60** | **5.10** |

Table 14: Evaluation metrics for T1ce modality completion on the BraTS 2020 dataset. To save place, SSIM values are given in e-1 order, MAE values are given in e-2 order, and MSE values are given in e-3 order. 'sota' refers to the original state-of-the-art methods without modifications, while 'ad.' refers to our adapted versions of these methods modified to handle semi-supervised scenarios.

| | BraTS 2020 | PSNR↑ | | | SSIM↑ | | | MAE↓ | | | MSE↓ | | |
|---|---|---|---|---|---|---|---|---|---|---|---|---|---|
| | T1ce | P100% | P80% | P50% | P100% | P80% | P50% | P100% | P80% | P50% | P100% | P80% | P50% |
| sota | MSDM† | **27.16** | 25.97 | 19.07 | **9.11** | 8.40 | 1.35 | **1.69** | 2.15 | 7.14 | 1.92 | 2.53 | 12.40 |
| | UMM-CSGM | 26.95 | 26.49 | 26.56 | 9.10 | 8.98 | 8.92 | 1.72 | 1.88 | 1.90 | 2.02 | 2.24 | 2.21 |
| ad. | MSDM†[N] | 27.06 | 24.51 | 16.94 | 8.87 | 6.99 | 0.60 | 1.83 | 2.68 | 10.10 | 1.97 | 3.54 | 20.22 |
| | UMM[N] | 26.96 | 27.14 | 26.68 | 8.92 | 8.88 | 8.80 | 1.81 | 1.78 | 1.89 | 2.02 | 1.93 | 2.15 |
| ours | MDD[RAND] | 26.82 | 27.14 | 26.92 | 8.73 | 8.59 | 8.92 | 1.90 | 1.85 | 1.76 | 2.08 | 1.93 | 2.03 |
| | MDD[NOISY] | 26.98 | 26.80 | 26.62 | 9.09 | 9.01 | **9.00** | 1.74 | 1.81 | 1.84 | 2.01 | 2.09 | 2.18 |
| | MDD | 27.05 | **27.57** | **27.28** | 8.94 | **9.09** | 8.92 | 1.81 | **1.63** | **1.73** | 1.97 | **1.75** | **1.87** |

Table 15: Evaluation metrics for T2 modality completion on the BraTS 2020 dataset. To save place, SSIM values are given in e-1 order, MAE values are given in e-2 order, and MSE values are given in e-3 order. 'sota' refers to the original state-of-the-art methods without modifications, while 'ad.' refers to our adapted versions of these methods modified to handle semi-supervised scenarios.

| | BraTS 2020 | PSNR↑ | | | SSIM↑ | | | MAE↓ | | | MSE↓ | | |
|---|---|---|---|---|---|---|---|---|---|---|---|---|---|
| | T2 | P100% | P80% | P50% | P100% | P80% | P50% | P100% | P80% | P50% | P100% | P80% | P50% |
| sota | MSDM† | 24.68 | 21.48 | 15.84 | 8.24 | 7.75 | 0.90 | 2.33 | 3.49 | 11.92 | 3.40 | 7.10 | 26.03 |
| | UMM-CSGM | 24.71 | 24.24 | 23.98 | 8.13 | 8.74 | 8.05 | 2.36 | 2.28 | 2.59 | 3.38 | 3.77 | 4.00 |
| ad. | MSDM†[N] | 25.25 | 20.89 | 12.32 | **8.91** | 7.01 | 0.29 | **2.11** | 3.82 | 16.26 | 2.99 | 8.14 | 58.60 |
| | UMM[N] | 24.73 | 24.84 | 24.04 | 8.82 | 8.87 | 8.61 | 2.24 | 2.23 | 2.46 | 3.37 | 3.28 | 3.95 |
| ours | MDD[RAND] | 24.86 | 24.54 | 24.29 | 8.64 | 8.46 | 8.61 | 2.20 | 2.34 | 2.35 | 3.26 | 3.52 | 3.73 |
| | MDD[NOISY] | 24.01 | 24.12 | 23.59 | 8.76 | 8.70 | 8.42 | 2.34 | 2.34 | 2.59 | 3.97 | 3.87 | 4.38 |
| | MDD | **25.38** | **25.72** | **24.79** | 8.75 | **8.94** | **8.77** | 2.12 | **1.98** | **2.23** | **2.90** | **2.68** | **3.32** |

## B.8 CelebAMask-HQ Additional Translation Results

We provide additional quantitative and qualitative results for (face, mask)→sketch translation in Fig. 21 and Tab. 18, (face, sketch)→mask in Fig. 22 and Tab. 19 translation and ()→(face, sketch, mask) generation in Fig. 23.

Table 16: Evaluation metrics for Flair modality completion on the BraTS 2020 dataset. To save place, SSIM values are given in e-1 order, MAE values are given in e-2 order, and MSE values are given in e-3 order. 'sota' refers to the original state-of-the-art methods without modifications, while 'ad.' refers to our adapted versions of these methods modified to handle semi-supervised scenarios.

| | BraTS 2020 | PSNR↑ | | | SSIM↑ | | | MAE↓ | | | MSE↓ | | |
|---|---|---|---|---|---|---|---|---|---|---|---|---|---|
| | Flair | P100% | P80% | P50% | P100% | P80% | P50% | P100% | P80% | P50% | P100% | P80% | P50% |
| sota | MSDM† | 22.80 | 18.83 | 17.76 | 8.72 | 7.85 | 2.29 | 2.85 | 5.01 | 7.24 | 5.25 | 13.10 | 16.74 |
| | UMM-CSGM | 22.81 | 22.60 | 22.06 | 8.71 | 8.60 | 8.33 | 2.85 | 3.00 | 3.32 | 5.24 | 5.49 | 6.22 |
| ad. | MSDM†[N] | 23.49 | 21.65 | 12.37 | 8.39 | 7.53 | 0.42 | 2.79 | 3.53 | 16.28 | 4.48 | 6.84 | 57.99 |
| | UMM[N] | 23.11 | 23.38 | 22.73 | 8.45 | 8.40 | 8.27 | 2.89 | 2.85 | 3.06 | 4.89 | 4.59 | 5.34 |
| ours | MDD[RAND] | **24.38** | 23.91 | 23.09 | 8.43 | 8.44 | 8.38 | 2.48 | 2.60 | 2.87 | **3.65** | 4.06 | 4.91 |
| | MDD[NOISY] | 22.83 | 22.86 | 22.16 | 8.66 | 8.66 | 8.56 | 2.90 | 2.90 | 3.16 | 5.21 | 5.18 | 6.07 |
| | MDD | 24.17 | **24.10** | **23.66** | **8.76** | **8.79** | **8.68** | **2.45** | **2.45** | **2.61** | 3.83 | **3.89** | **4.30** |

Table 17: Evaluation metrics for Mask modality completion on the BraTS 2020 dataset. 'sota' refers to the original state-of-the-art methods without modifications, while 'ad.' refers to our adapted versions of these methods modified to handle semi-supervised scenarios.

| | BraTS 2020 | Jaccard↑ | | |
|---|---|---|---|---|
| | Segmentation | P100% | P80% | P50% |
| sota | MSDM† | 0.056 | 0.049 | 0.000 |
| | UMM-CSGM | 0.755 | 0.072 | 0.002 |
| ad. | MSDM†[N] | 0.055 | 0.066 | 0.071 |
| | UMM[N] | 0.727 | 0.770 | 0.713 |
| ours | MDD[RAND] | 0.752 | 0.734 | 0.695 |
| | MDD[NOISY] | 0.061 | 0.057 | 0.059 |
| | MDD | **0.788** | **0.788** | **0.763** |

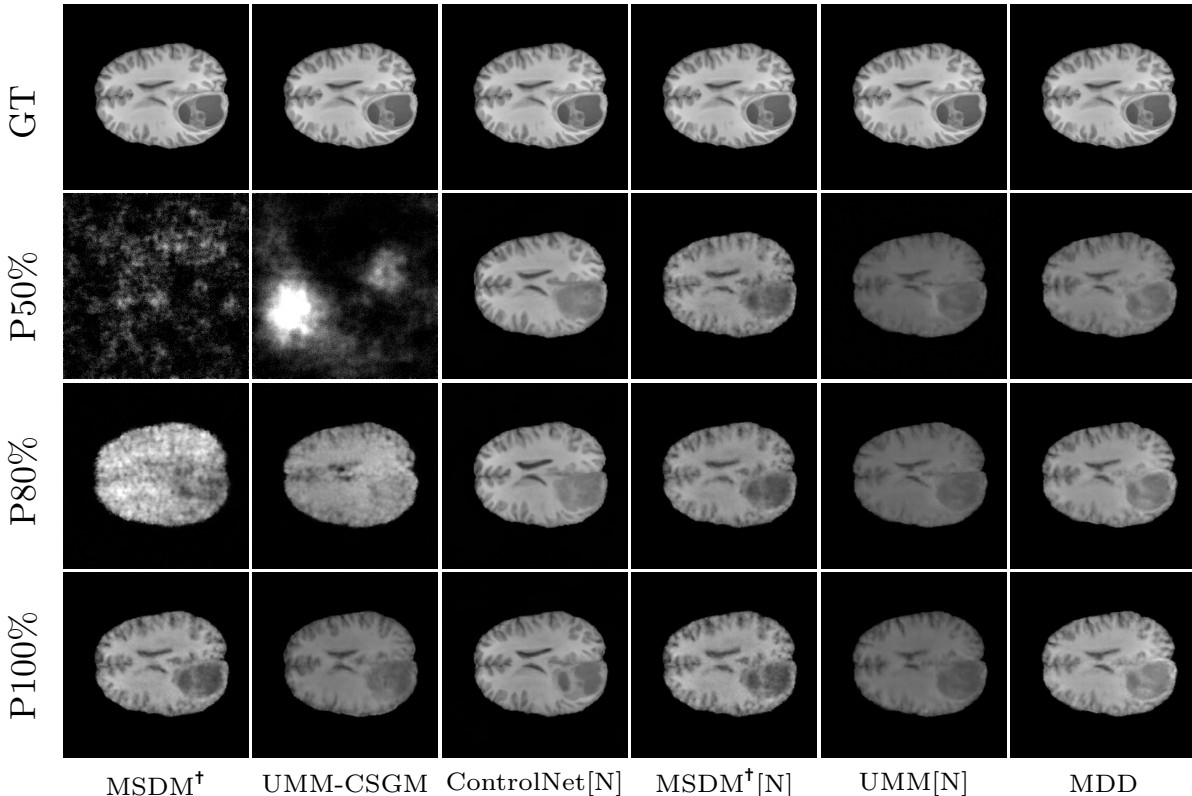

MSDM†        UMM-CSGM    ControlNet[N]    MSDM†[N]       UMM[N]        MDD

Figure 16: Examples of generated T1 scan given all remaining modalities, and the corresponding GT for different levels of supervision.

The diversity of face generation is evaluated in different settings: (sketch, mask)→face generation Fig. 24 and mask→(face, sketch) Fig. 25 where we calculate the Diversity Score over the generated faces Tab. 20.

The translation results for sketch and mask generation are consistent with those presented in the main paper, and on the BraTS 2020 dataset, MDD models perform strongly. For the semantic segmentation task, MSDM†[N] performs better than on the BraTS 2020 dataset while still achieving the lowest Jaccard. This can be explained by the fact that a correct "mean" prediction is easier for the CelebAMask-HQ dataset, since the segmentations for different images are similar (the faces are centered, often have the same angles *etc.*).

We found that the noisy condition model, MSDM†[N], has the best diversity score for the generated faces but the worst generation quality. Using a noisy condition allows the model to avoid using it at the beginning

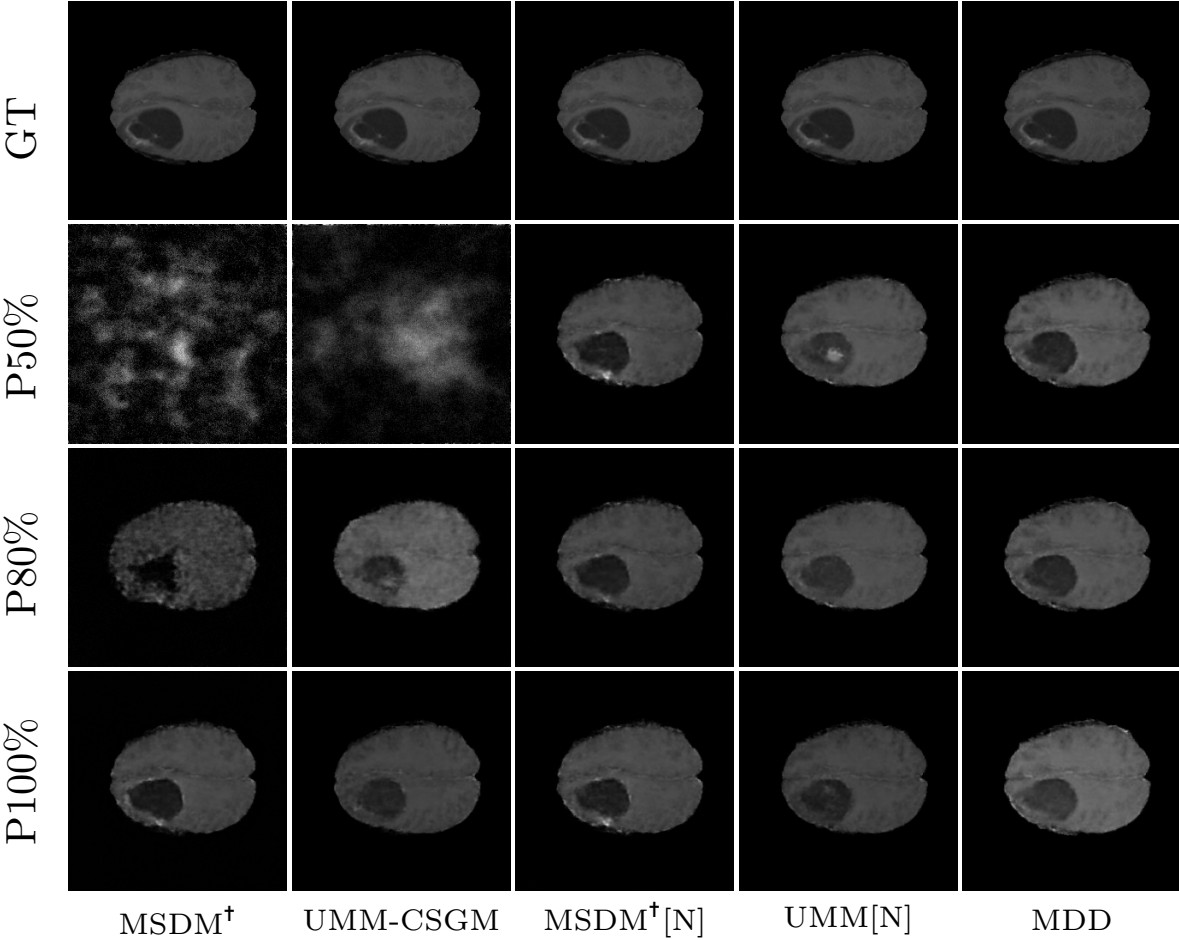

Figure 17: Examples of generated T1ce scan given all remaining modalities, and the corresponding GT for different levels of supervision.

of the generation process and use a noisy condition later in the diffusion process. This allows the generated faces to drift further away from the condition, producing more diverse images. In the case of MSDM$^\dagger$[N], while it has the higher diversity, it also has the worst generation quality according to Tab. 3.

Table 18: Evaluation metrics for (Face, Mask)→Sketch on the CelebAMask-HQ dataset. 'ad.' refers to our adapted versions of state-of-the-art methods modified to handle semi-supervised scenarios.

| | CelebAMask-HQ | PSNR↑ | | SSIM↑ | |
|---|---|---|---|---|---|
| | Sketch | 100% | 0% | 100% | 0% |
| ad. | MSDM$^\dagger$[N] | 17.298 | 16.652 | 0.3875 | 0.3739 |
| | UMM[N] | 18.576 | 16.872 | 0.4371 | 0.3897 |
| ours | MDD[RAND] | 17.526 | 17.071 | 0.3903 | 0.3715 |
| | MDD | **20.402** | **18.259** | **0.5055** | **0.4219** |

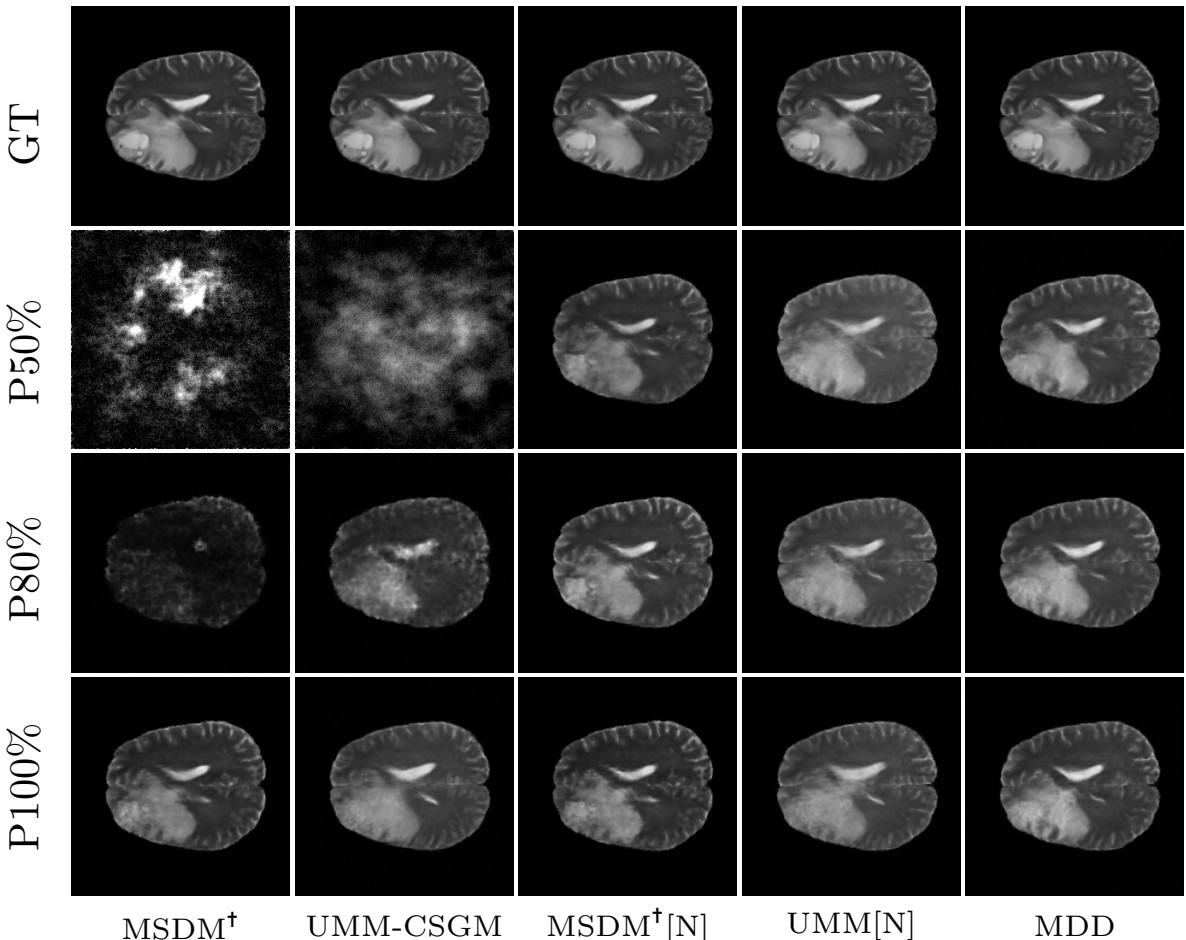

Figure 18: Examples of generated T2 scan given all remaining modalities, and the corresponding GT for different levels of supervision.

Table 19: Evaluation metrics for (Face, Sketch)→Mask on the CelebAMask-HQ dataset. 'ad.' refers to our adapted versions of state-of-the-art methods modified to handle semi-supervised scenarios.

|  | CelebAMask-HQ | Jaccard↑ | |
|---|---|---|---|
|  | Mask | 100% | 0% |
| ad. | MSDM†[N] | 0.4476 | 0.4731 |
| ad. | UMM[N] | 0.7378 | 0.7131 |
| ours | MDD[RAND] | 0.6978 | 0.6704 |
| ours | MDD | **0.7536** | **0.7133** |

Table 20: Diversity of Face generation metrics for (Sketch, Mask)→Face generation and Mask→(Face, Sketch) on the CelebAMask-HQ dataset. 'ad.' refers to our adapted versions of state-of-the-art methods modified to handle semi-supervised scenarios.

|  | CelebAMask-HQ | (S,M)→(F) DS↑ | | (M)→(S,F) DS↑ | |
|---|---|---|---|---|---|
|  | Face Diversity | 100% | 0% | 100% | 0% |
| ad. | MSDM†[N] | 19.883 | 18.625 | 19.896 | 18.726 |
| ad. | UMM[N] | 19.668 | 18.213 | 20.305 | 18.564 |
| ours | MDD[RAND] | 16.518 | 16.028 | 17.502 | 16.614 |
| ours | MDD | 19.524 | 15.842 | 19.177 | 15.786 |

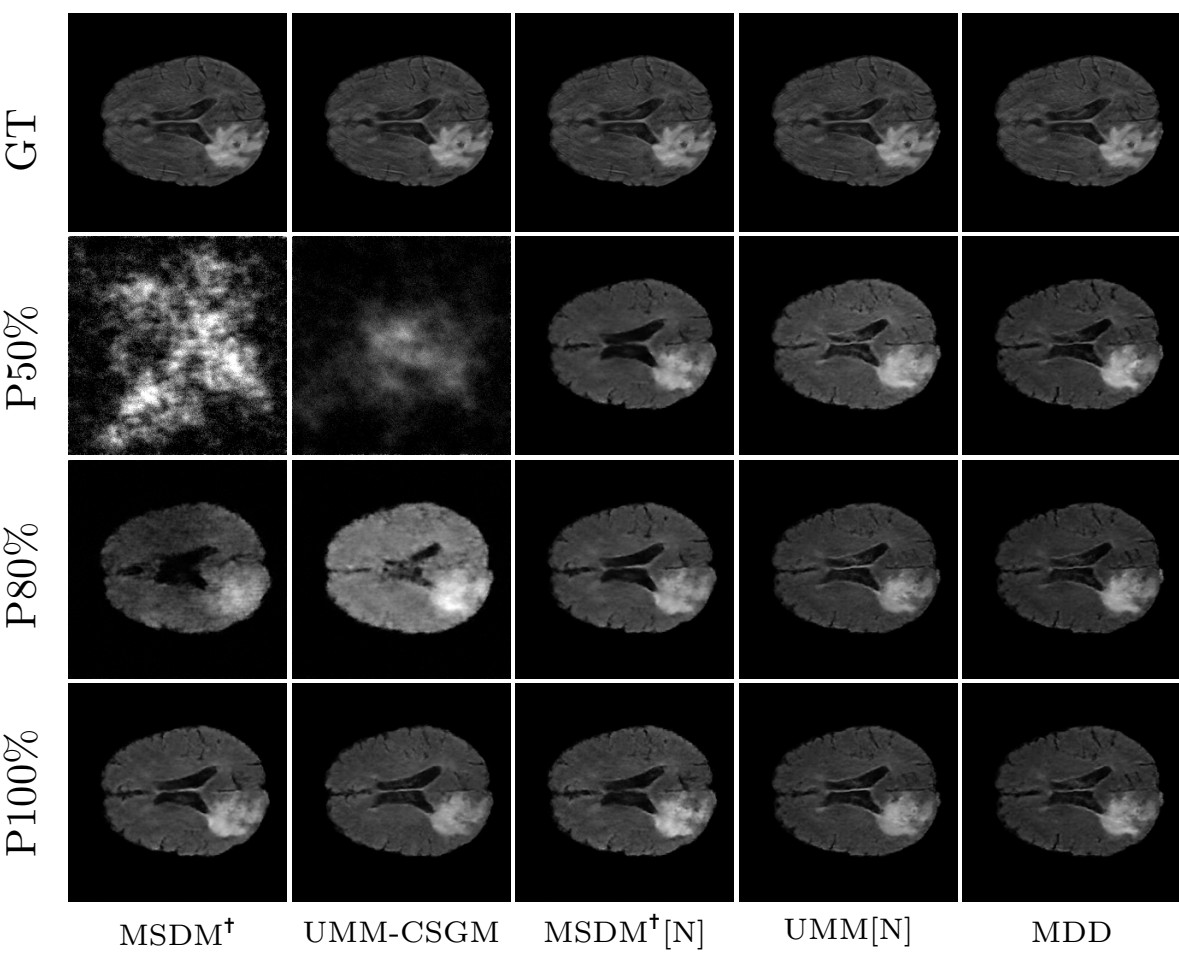

Figure 19: Examples of generated Flair scan given all remaining modalities, and the corresponding GT for different levels of supervision.

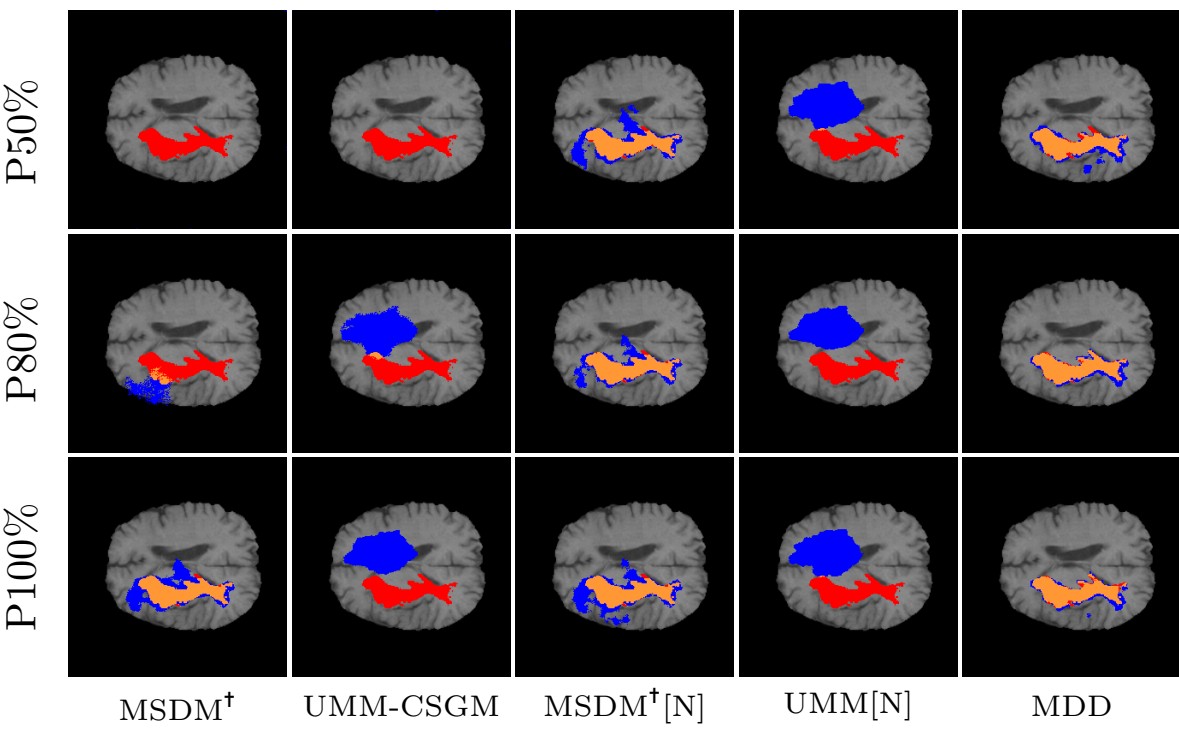

Figure 20: Generated segmentation given all remaining modalities. Orange represents true positives, red false negatives, and blue false positives.

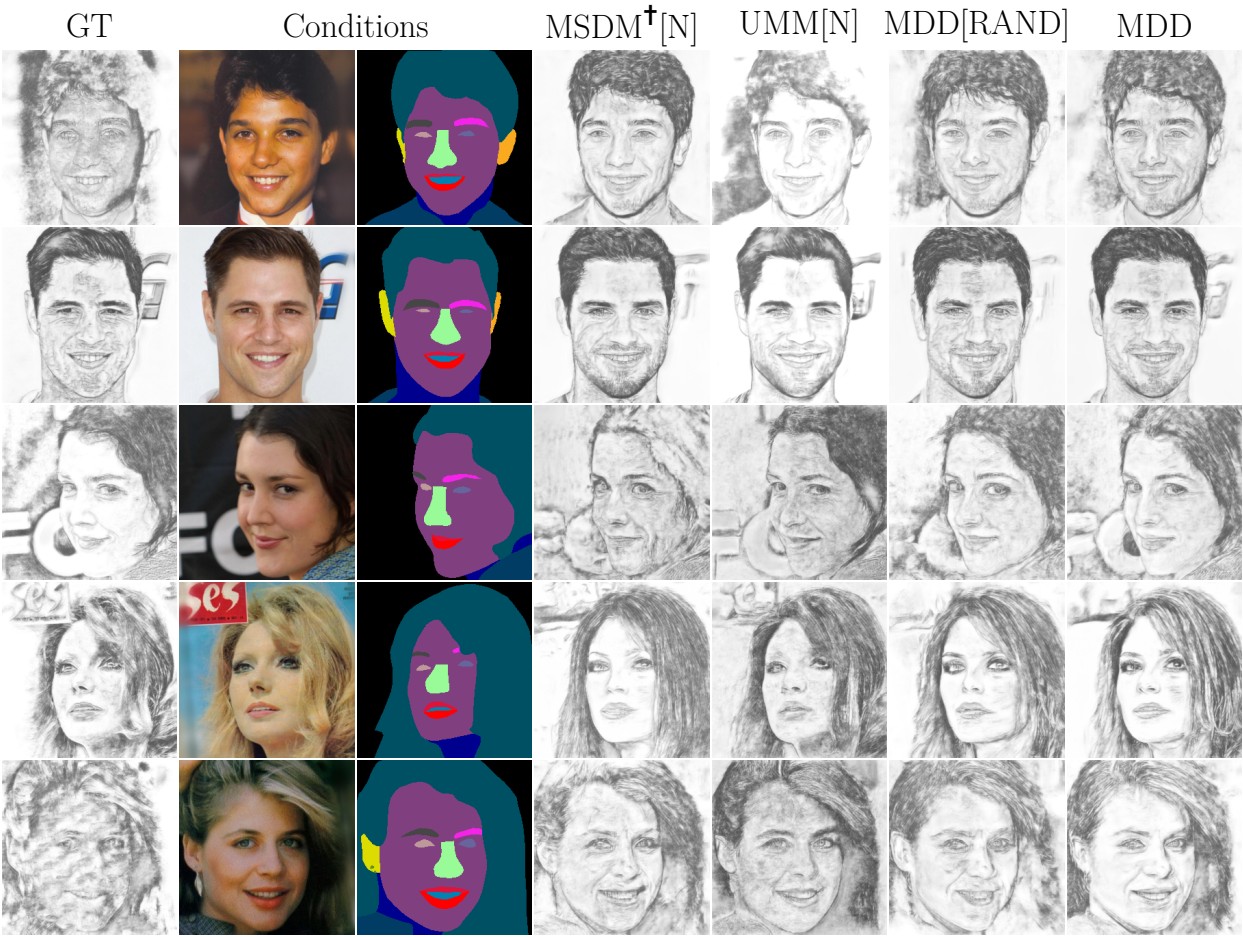

Figure 21: Examples of generated sketches given a face and a mask, and the corresponding GT for N=0%.

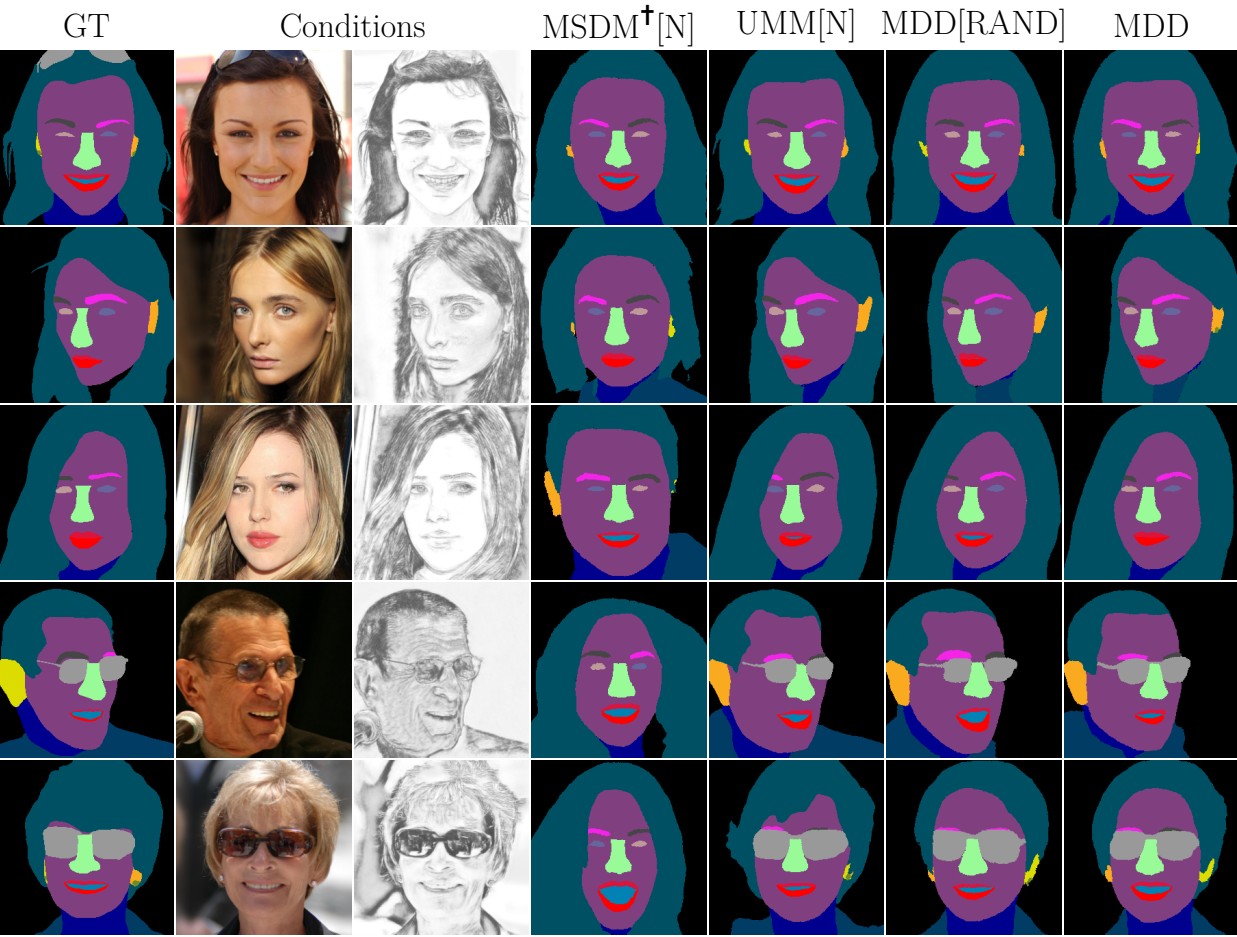

Figure 22: Examples of generated masks given a face and a sketch, and the corresponding GT for N=0%.

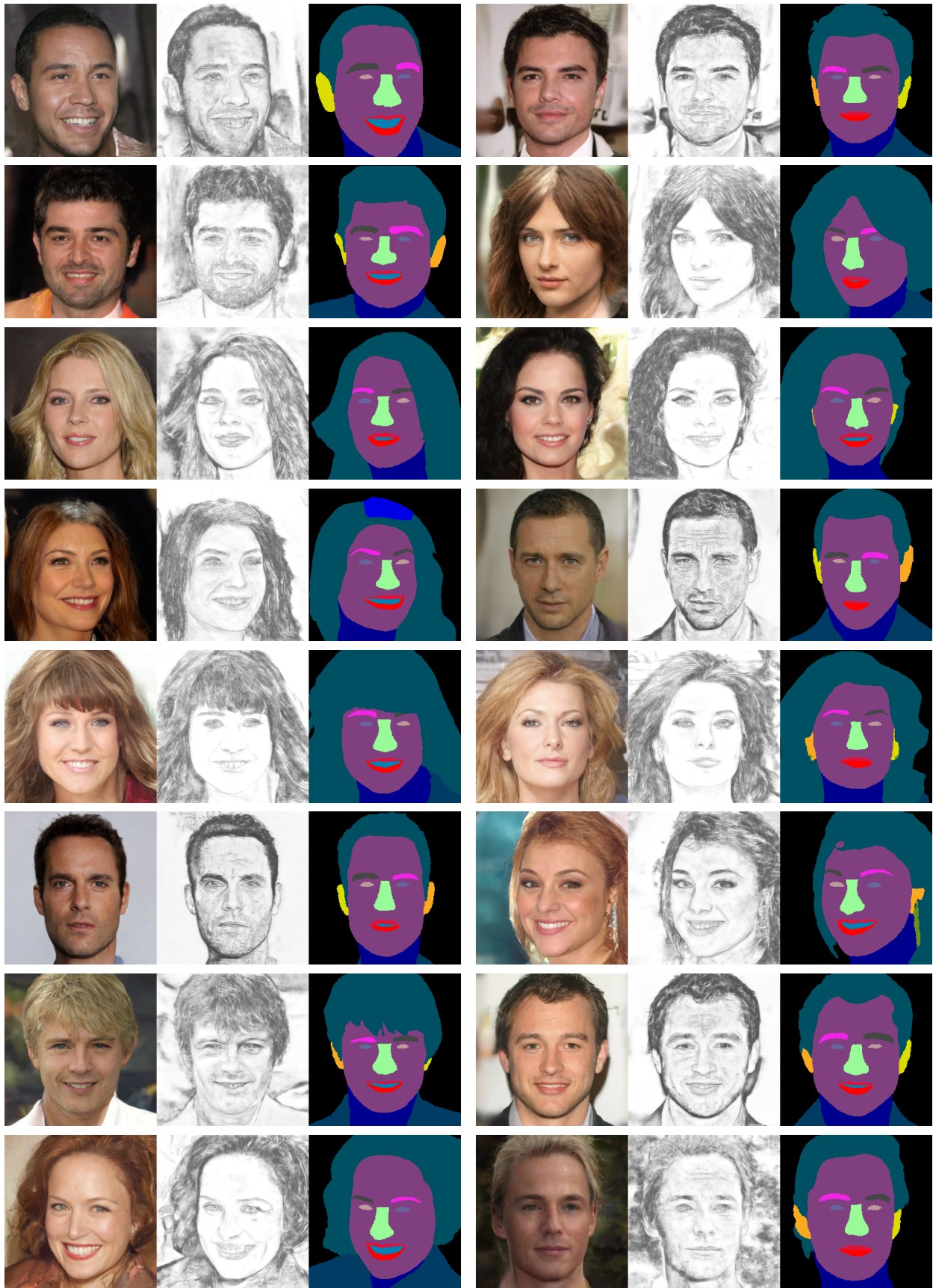

Figure 23: Examples of unconditional generation ()→(Face,Sketch,Mask) for MDD N=100%.

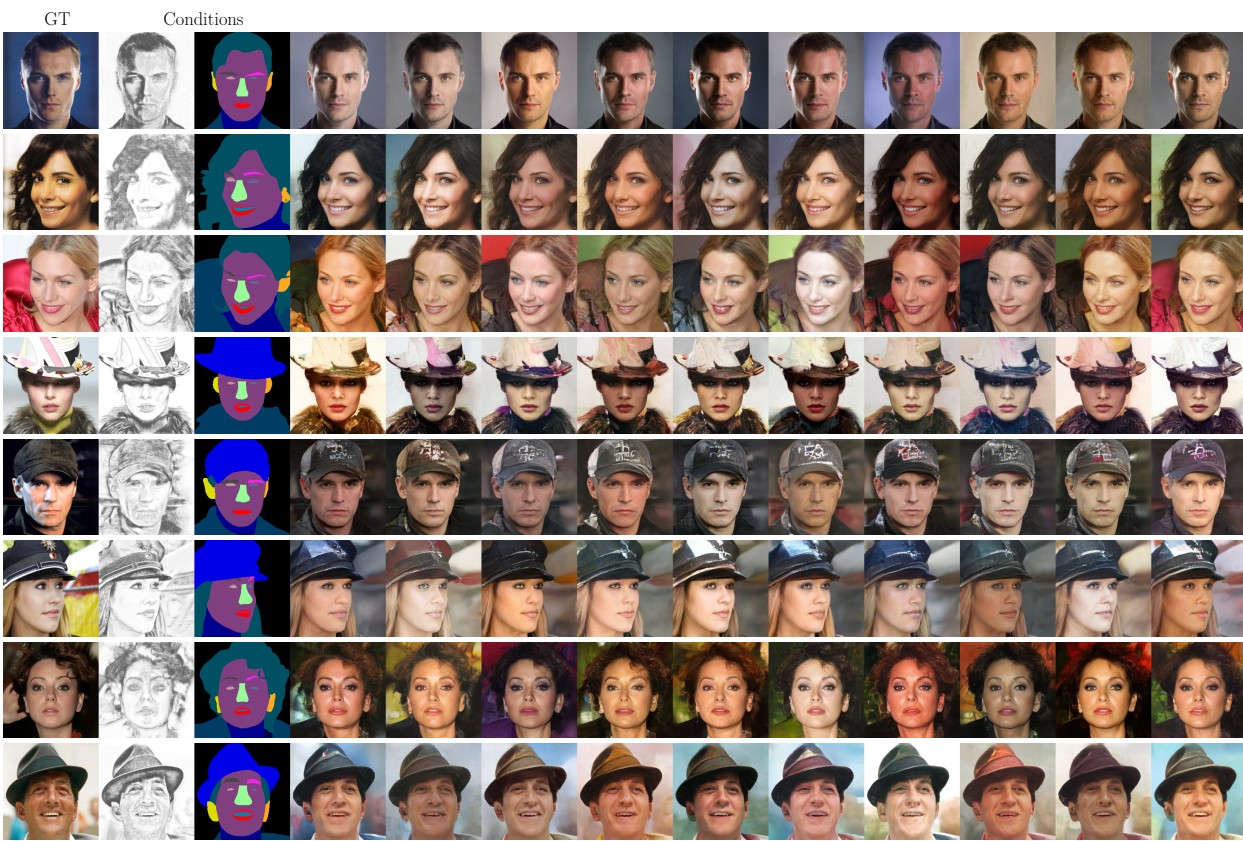

Figure 24: Examples of diversity face generation (sketch,mask)→face for MDD N=100%.

GT    Condition

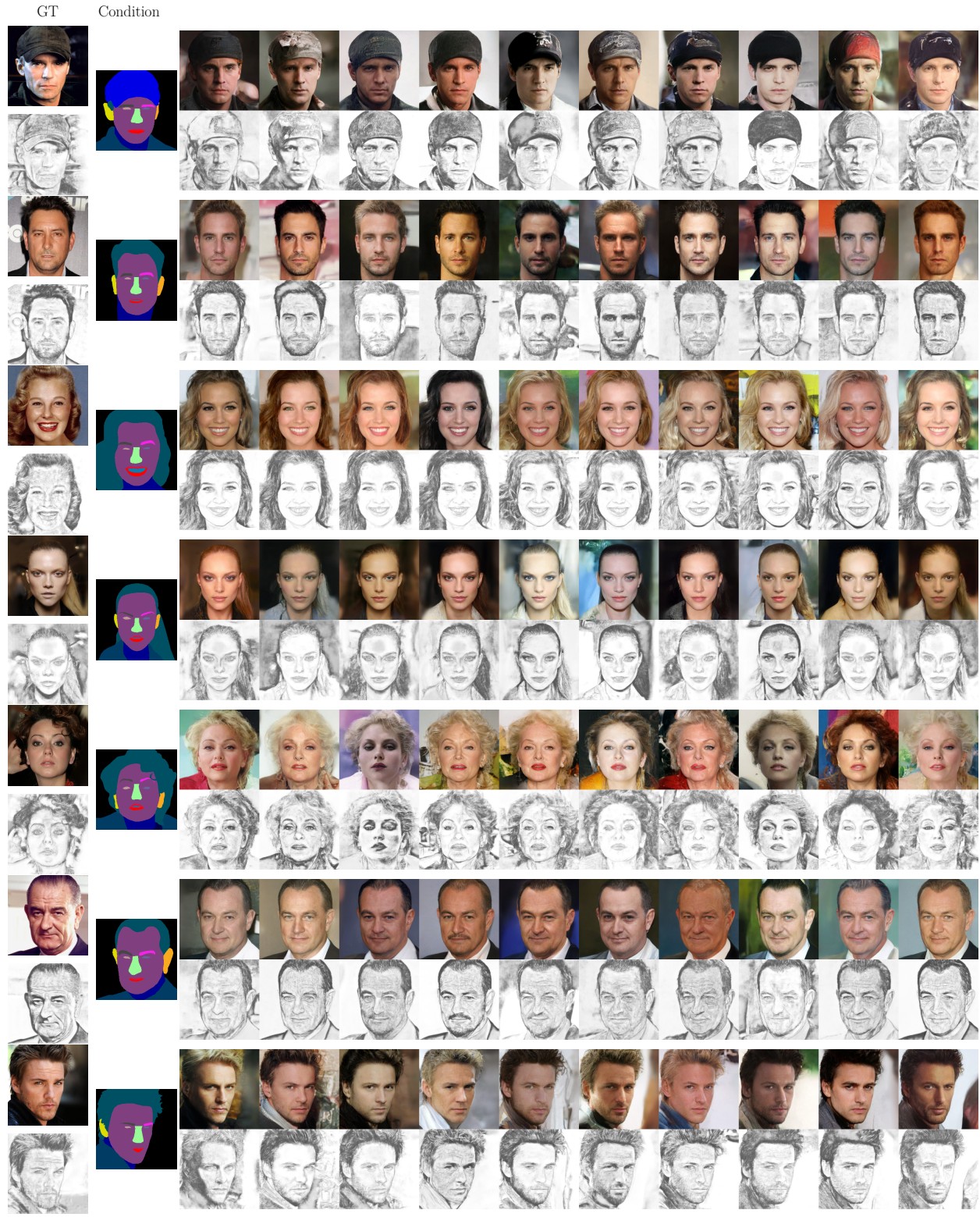

Figure 25: Examples of diversity face and sketch generation Mask→(Face, Sketch) for MDD N=100%.

