# OpenReview forum: "Multiple Noises in Diffusion Model for Semi-Supervised Multi-Domain Translation"
_TMLR — Accepted by TMLR_

### Review · Reviewer_uQXo · 2025-05-19

**Summary Of Contributions:**

This paper introduces an interesting diffusion-based method (MDD) for multi-domain translation that enables mappings between arbitrary combinations of L domains (i.e., $2^{L}$ possible functions) using a single model. Unlike existing methods that train separate models for specific source-to-target translations, MDD can handle two parts: (1) reconstructing missing views (i.e., absent data for specific domains) for new data points, and (2) supporting semi-supervised learning with arbitrary supervision configurations. Missing views, such as an unavailable segmentation mask in a dataset with face images, sketches, and masks, are treated as maximally noisy inputs (i.e., t=T). The authors conduct a wide range of experiments cover fully supervised, semi-supervised, and bridge settings (where certain domain pairs are never seen together), comparing MDD against state-of-the-art methods, and include a detailed ablation study.

**Audience:**

Yes

**Claims And Evidence:**

Yes

**Requested Changes:**

1. Add a brief example in Section 3.2 to clarify noise level masking for missing views, e.g., for CelebAMask-HQ with a missing segmentation mask. For example, consider a data point with domains $D_1$, $D_2$, and $D_3$, where $D_3$ is missing.
(1) Show the supervision mask $m = [1, 1, 0]$, sampled noise levels $\mathcal{T}_{\text{sup}} = [200, 500, 300]$, and how $\mathcal{T} = [200, 500, T]$ is computed with $T = 1000$; (2) explain how the input for $D_3$ is replaced with Gaussian noise and used in training (Algorithm 1, line 5).

2. The evaluation shows that MDD struggles in the BL3NDT bridge setting (MAE of 1.199 v.s. 0.316 at 100% supervision), where missing views are reconstructed via intermediate domains. This suggests potential reconstruction conflicts due to error propagation or complex inter-domain dependencies. Add a paragraph in the discussion section to address how MDD could mitigate such conflicts.

3. MDD uses a fixed clean condition ($\phi(t) = 0$) during generation, which ensures fidelity but may limit diversity in generative tasks.  Please consider adding a discussion in Section 3.3 on the choice of $\phi(t)$ and propose an adaptive strategy based on task type.
For example, for discriminative tasks (e.g., segmentation), use $\phi(t) = 0$ for precision; while for generative tasks, use a noisy condition (e.g., $\phi_\gamma(t)$ with $\gamma = 0.7$ to increase diversity. Also add a table or figure in the results comparing $\phi(t) = 0$ and an adaptive $\phi_\gamma(t)$ on CelebAMask-HQ, reporting LPIPS and Diversity Score.

4. MDD’s performance relies on uniform sampling of noise levels for available domains and a linear noise schedule. The BL3NDT bridge setting suggests sensitivity to noise in intermediate domains, which raises concerns about reconstruction stability for missing views. Sp, please add a paragraph or subsection in the appendix analyzing noise schedule sensitivity. For example, evaluate MDD with alternative schedules (e.g., cosine schedule, as used in some diffusion models), report the impact on BL3NDT bridge setting MAE or BraTS 2020 PSNR at P50% supervision, and discuss implications for optimizing $\mathcal{T}_{\text{sup}}$ sampling (e.g., prioritizing lower noise levels for more informed domains).

5. Please include a computational analysis and compare the proposed MDD with the existing methods.

**Strengths And Weaknesses:**

### Strengths

1. This paper introduces a novel method that assigns independent noise levels to each domain within a noise vector, with missing views set to $t^{(i)} = T$ (white noise). The proposed method elegantly handles missing data in semi-supervised settings by treating absent views as fully noised inputs, outperforming methods like MSDM that use uniform noise levels.

2. By learning the joint data distribution across all domains, MDD supports translations between any domain subset without requiring separate models, a significant improvement over methods like Pix2Pix or CycleGAN. This scalability is evident in its ability to handle $2^L$ mappings with only one model.

3. The supervision mask and noise level masking allow MDD to naturally adapt to semi-supervised settings. For missing views, by setting $t^{(i)} = T$ enables the model to reconstruct absent domains using available ones, with promising performance in low-supervision settings.

4. The paper evalutes MDD across diverse supervision levels, datasets, and baselines. Additional comparisons with ControlNet and noise strategy explorations $\phi(t)$ in the appendix strengthen the analysis.

### Weaknesses

1. The proposed MDD modified U-Net with duplicated encoders/decoders and an aggregation network, which increases computational cost compared to other methods. Training for 2000 epochs on CelebAMask-HQ and requiring large batch sizes (e.g., 256 for BL3NDT) may limit scalability, a common issue for diffusion models but might be amplified by multi-domain settings.

2. For generative tasks (e.g., face generation from sketches), multiple valid outputs are possible, but MDD’s default clean condition ($\phi(t) = 0$) may limit diversity. The appendix explores alternative $\phi(t)$ strategies, but the main results lack an adaptive method, potentially causing conflicts in tasks requiring balanced fidelity and diversity

3. The uniform sampling of noise levels for available domains and fixed at $\phi(t) = 0$ for generation may not be optimal for all tasks. The BL3NDT bridge setting shows sensitivity to noise in intermediate domains, and the lack of analysis on noise schedule robustness could affect reconstruction stability, especially for missing views.

---

> ### Author Response · Authors · 2025-07-09
>
> ### Weaknesses
>
> > The proposed MDD modified U-Net with duplicated encoders/decoders and an aggregation network, which increases computational cost compared to other methods. Training for 2000 epochs on CelebAMask-HQ and requiring large batch sizes (e.g., 256 for BL3NDT) may limit scalability, a common issue for diffusion models but might be amplified by multi-domain settings
>
> All multi-domain translation models evaluated in this paper (MDD, MSDM, UMM-CSGM) employ identical encoder/decoder architectures and aggregation network configurations. To ensure fair comparison, we implemented the same architectural framework across all methods, resulting in equivalent computational costs during inference.
>
> Regarding the batch size for BL3NDT, we note that the referenced value of 256 was incorrect in our original manuscript. The actual batch size used was 128, and we have corrected this error in the revised version.
>
> In response to Reviewer Uodi's suggestion, we have expanded Section 5 to include a discussion on potential scaling strategies for MDD when handling a substantial number of domains, addressing computational efficiency considerations for large-scale multi-domain translation tasks.
>
> > The uniform sampling of noise levels for available domains and fixed at $\phi(t)$
>  for generation may not be optimal for all tasks. The BL3NDT bridge setting shows sensitivity to noise in intermediate domains, and the lack of analysis on noise schedule robustness could affect reconstruction stability, especially for missing views.
>
> Indeed, we acknowledge that specific $\phi(t)$ functions might be more effective for certain tasks, depending on factors such as step count, diffusion model architecture, and the modalities involved. The selection of optimal schedulers for classical diffusion models is already highly empirical in nature [EDM].
>
> Designing an adaptive $\phi(t)$ function that comprehensively addresses these diverse factors is non-trivial and beyond the scope of the current work - such an investigation would warrant dedicated research. Future work could build upon approaches similar to those in Spatial Reasoning with Denoising Models [SRM] to adaptively determine appropriate time steps for each domain.
>
> Importantly, our proposed MDD consistently outperforms all other considered baselines, indicating that while our proposed $\phi(t)$ may not be optimal for every scenario, it serves as a reliable default strategy with strong empirical performance across the evaluated tasks.
>
> ### Requested Changes:
> > 1. Add a brief example in Section 3.2 to clarify noise level masking for missing views, e.g., for CelebAMask-HQ with a missing segmentation mask. For example, consider a data point with domains $D_1$, $D_2$, and $D_3$, where $D_3$ is missing. (1) Show the supervision mask $m=[1,1,0]$, sampled noise levels $\tau_\text{sup} = [200,500,T]$, and how $\tau=[200,500,T]$ is computed with $T=1000$; (2) explain how the input for $D_3$ is replaced with Gaussian noise and used in training (Algorithm 1, line 5).
>
> We thank the reviewer for this suggestion. To address the clarity concerns regarding noise level masking for missing views, we have enhanced our presentation in several ways:
>
> First, we have modified Figure 4 "Training procedure of MDD" to explicitly illustrate the training procedure with concrete time values and supervision masks. This visualization now clearly demonstrates how these elements interact during model training.
>
> Second, we have refined Section 3.3 "Noise Modeling for Semi-Supervised Multi-Domain Translation" to provide a more comprehensive explanation of the components referenced in point (1), including the supervision mask formulation and noise level computation.
>
> For point (2), we have incorporated this clarification both in the revised Figure 4 "Training procedure of MDD" and by adding a formal equation in Section 3.3. Additionally, we have expanded Algorithm 1 (line 11) to explicitly show how inputs for missing domains are replaced with Gaussian noise during the training process.
>
> These supplementary explanations should provide greater explanation regarding how MDD handle partially observed multi-domain data during training and generation.
>
> > 2. The evaluation shows that MDD struggles in the BL3NDT bridge setting (MAE of 1.199 v.s. 0.316 at 100% supervision), where missing views are reconstructed via intermediate domains. This suggests potential reconstruction conflicts due to error propagation or complex inter-domain dependencies. Add a paragraph in the discussion section to address how MDD could mitigate such conflicts.
>
> We have incorporated an expanded discussion addressing the BL3NDT Bridge domain translation challenges at the conclusion of Section 4.3 "BL3NDT Results". This additional analysis examines the performance discrepancies observed in bridge settings and proposes potential approaches for mitigating reconstruction conflicts and error propagation in complex inter-domain dependencies.

---

> ### Author Response · Authors · 2025-07-09
>
> > 3. MDD uses a fixed clean condition ($\phi(t) = 0$) during generation, which ensures fidelity but may limit diversity in generative tasks. Please consider adding a discussion in Section 3.3 on the choice of $\phi(t)$ and propose an adaptive strategy based on task type. For example, for discriminative tasks (e.g., segmentation), use $\phi(t) = 0$ for precision; while for generative tasks, use a noisy condition (e.g., $\phi_\gamma(t)$ with $\gamma = 0.7$ to increase diversity. Also add a table or figure in the results comparing $\phi(t) = 0$ and an adaptive $\phi_\gamma(t)$ on CelebAMask-HQ, reporting LPIPS and Diversity Score.
>
> Thank you for your feedback. We have addressed this valuable suggestion by:
>
> 1. Adding a comprehensive discussion at the end of Section 3.3 regarding the selection of appropriate $\phi$ strategies based on task requirements.
>
> This addition provides important insights into how different noise strategies can be optimized for specific task requirements, whether prioritizing precision (for discriminative tasks) or diversity (for generative tasks).
>
> 2. Including a new experimental analysis in Section B.3 "Sampling Strategy Impact on Diversity," which specifically examines:
>    - The trade-off between image diversity and condition fidelity on CelebAHQ
>    - A comparison of different $\phi$ strategies, including $\phi(t) = 0$ and adaptive $\phi_\gamma(t)$
>    - Quantitative evaluation using LPIPS and Diversity Score metrics
>
> > 4. MDD’s performance relies on uniform sampling of noise levels for available domains and a linear noise schedule. The BL3NDT bridge setting suggests sensitivity to noise in intermediate domains, which raises concerns about reconstruction stability for missing views. Sp, please add a paragraph or subsection in the appendix analyzing noise schedule sensitivity. For example, evaluate MDD with alternative schedules (e.g., cosine schedule, as used in some diffusion models), report the impact on BL3NDT bridge setting MAE or BraTS 2020 PSNR at P50% supervision, and discuss implications for optimizing $\tau_\text{sup}$ sampling (e.g., prioritizing lower noise levels for more informed domains).
>
> We thank the reviewer for this valuable suggestion. As recommended, we have included additional experiments using alternative schedulers (specifically cosine-based) in Section B.4 "Sensitivity to Noise Schedulers".
> Across all our evaluations using the cosine-based scheduler, we found consistent improvements compared to our previously reported results with the linear scheduler.
> These findings highlight how MDD could benefit from recent advancements in the diffusion model literature, suggesting that noise schedule optimization represents a promising direction for further enhancing the performance of our approach.
>
> > 5. Please include a computational analysis and compare the proposed MDD with the existing methods.
>
> We have added additional experiments on MDD's computational performance in Section B.5 "MDD Computational Analysis".
> Our analysis demonstrates that MDD enables faster translation speeds in multi-domain translation settings compared to methods that perform translation of one domain at a time.
> This computational advantage further supports MDD's practical utility for real-world applications requiring simultaneous multi-domain translations.
>
> [SRM] Wewer, Christopher, et al. "Spatial Reasoning with Denoising Models." International Conference on Machine Learning (2025)
>
> [EDM] Karras, Tero, et al. "Elucidating the design space of diffusion-based generative models." Advances in neural information processing systems 35 (2022)

---

> ### Comment · Reviewer_uQXo · 2025-08-05
>
> Thank you the authors for the detailed point-by-point responses. All my concerns are properly addressed. I am happy to recommend acceptance.

---

### Review · Reviewer_Uodi · 2025-05-29

**Summary Of Contributions:**

This paper proposes Multi-Domain Diffusion (MDD), a novel framework for semi-supervised multi-domain translation based on diffusion models. The method introduces:

1. A multi-noise-level formulation where each domain has an independent noise level during the diffusion process.

2. An ability to handle missing views naturally during training and inference by treating them as high-noise representations.

3. A flexible approach to semi-supervised learning across arbitrary domain configurations without requiring fixed input-output paths.

4. Extensive experiments on three datasets (BL3NDT, CelebAMask-HQ, and BraTS 2020) showing superior performance over adapted and state-of-the-art baselines.

**Audience:**

Yes

**Claims And Evidence:**

Yes

**Requested Changes:**

1. Clarify fusion strategy scalability in Sec. 5: Provide a brief discussion or suggestion on how future work could address architectural scaling with >5 domains.This is important because MDD's applicability to large-scale multi-modal tasks (e.g., video, multi-sensor fusion) depends on this.

2. Better motivate the clean vs. noisy condition choice in Sec. 3.2 and App. B.1: While clean condition performs better, provide more theoretical or intuitive reasoning—beyond empirical evidence—about why noisy conditioning degrades generation significantly. Clarify whether clean vs. noisy conditioning generalises across tasks with different generative ambiguities (e.g., deterministic vs. stochastic mappings).

**Strengths And Weaknesses:**

Strengths：

1.  The whole idea is pretty novel. The paper proposes a creative formulation that treats each domain with its own noise level, effectively modeling missing data and enabling flexible generation.

2. The method is flexible. MDD supports both full and semi-supervised settings, arbitrary conditional configurations, and can operate with varying numbers of available modalities at inference.

3. Extensive experiments demonstrate its effectiveness.

Weakness:

1. As acknowledged by the authors, the method's scalability with a large number of domains may be limited due to potential bottlenecks in multi-modal fusion within a single latent vector. This needs more further explanation on solutions.

2. While a U-Net backbone is used, more comparative discussion on architectural design choices (e.g., handling of skip connections, time embedding for multi-domain setting) would help clarify contributions over baseline architectures.

---

> ### Author Response · Authors · 2025-07-09
>
> ### Weakness
>
> > 2. While a U-Net backbone is used, more comparative discussion on architectural design choices (e.g., handling of skip connections, time embedding for multi-domain setting) would help clarify contributions over baseline architectures.
>
> To further clarify MDD's architectural design, we have added Figure 11 in Section A.4 "Model Architecture and Adaptation to Multiple Domains," which explicitly illustrates our UNet design choices and multi-domain adaptations.
>
> It is important to note that all multi-domain translation models evaluated in our experiments (MDD, UMM-CSGM, and MSDM) utilize the same underlying architecture, ensuring fair comparisons focused on the algorithmic contributions rather than architectural advantages.
>
> ### Requested Changes:
>
> > 1. Clarify fusion strategy scalability in Sec. 5: Provide a brief discussion or suggestion on how future work could address architectural scaling with >5 domains.This is important because MDD's applicability to large-scale multi-modal tasks (e.g., video, multi-sensor fusion) depends on this.
>
> We have included an additional discussion at the end of Section 5 "Limitations and Future Directions" introducing possible future research directions that could allow MDD to scale to a larger number of domains. We have also delineated potential limitations that would need to be overcome to achieve further scaling, particularly for applications involving heterogeneous modalities or high-dimensional data streams.
>
> > 2. Better motivate the clean vs. noisy condition choice in Sec. 3.2 and App. B.1: While clean condition performs better, provide more theoretical or intuitive reasoning—beyond empirical evidence—about why noisy conditioning degrades generation significantly. Clarify whether clean vs. noisy conditioning generalises across tasks with different generative ambiguities (e.g., deterministic vs. stochastic mappings).
>
> We thank the reviewer for pointing out that the intuition behind the motivation for clean vs. noisy conditioning was lacking.
> We have further clarified why and how clean conditioning performs better than noisy conditioning by adding both intuitive explanations and theoretical insights into the generative process in Section 3.2 "Existing Issues With Noisy Conditional Diffusion Models." We have also added two new figures: Figure 2 "Schematic illustration of Noisy Conditional Diffusion Models vs MDD" and Figure 3 "Illustration comparing (top) noisy condition with MSDM (Mariani et al., 2024) and (bottom) clean condition with MDD."
>
> In response to Reviewer uQXo's suggestion, we have expanded Section 3.3 "Noise Modeling for Semi-Supervised Multi-Domain Translation" to include additional details on the choice of $\phi$ for generative or discriminative tasks. This clarification helps establish the theoretical foundation for when different noise modeling strategies are most appropriate and how they impact the model's behavior across diverse translation scenarios.
>
> To further study the difference between clean and noisy conditioning for deterministic and stochastic mappings, we have added new experimental analysis in Section B.3 "Sampling Strategy Impact on Diversity," which specifically examines:
>    - The trade-off between image diversity and condition fidelity on CelebAHQ
>    - A comparison of different $\phi$ strategies
>    - Quantitative evaluation using LPIPS and Diversity Score metrics
>
> We found that MDD can achieve a controllable trade-off in quality vs. diversity, allowing it to match the behavior of noisy conditioning models when necessary while retaining the ability to produce more accurate outputs when fidelity is prioritized.
>
> These additions provide visual and conceptual clarity on the advantages of our method over noisy conditioning across different types of domain translation tasks, including those with varying degrees of generative ambiguity.

---

> > ### Comment · Reviewer_Uodi · 2025-08-07
> >
> > I have read the author's rebuttal and discussion with other reviewers. I believe my concerns are addressed greatly. I am glad to recommend to accept this paper.

---

### Review · Reviewer_BEHa · 2025-07-08

**Summary Of Contributions:**

This paper proposes a new strategy for multi-source domain translation, a problem where one wants to translate inputs (i.e., map) between different domains. The authors do so through diffusin models, semi-supervision and masking, i.e., they produce multiple views of a ground-truth data point through masking, and the diffusion model must reconstruct the masked input. The authors demonstrate, on various image benchmarks, the superiority of their method with respect other diffusion-based domain translation models.

**Audience:**

Yes

**Claims And Evidence:**

Yes

**Requested Changes:**

While I think this paper would benefit from a deeper theoretical analysis, in its current format it is ready for publication at TMLR, especially since:

1. The empirical validation confirms the paper claims,
2. The paper fits the scope of the journal.

I have only one minor requested change. In section 3.3, the authors mention that they apply a mask $x_{supervised} = x \odot m$ and $x_{unavailable} = x \odot (1-m)$. Here, I am supposing $C = A \odot B \iff C_{ij} = A_{ij}B_{ij}$, but I would like the authors to: 1) confirm this, 2) include, in the main text, a definition for this operation.

**Strengths And Weaknesses:**

Positive points:
+ The method is based on a simple idea (masking)
+ The experimental section is comprehensive
+ The proposed strategy increases performance over baselines/SOTA
+ Limitation section is good

Negative points:
- No theoretical justification or analysis of the method
- Some unclear notation (minor)

---

> ### Author Response · Authors · 2025-07-09
>
> We thank the reviewer for the careful proofreading and are pleased that you found our work valuable.
>
> > No theoretical justification or analysis of the method
>
> Multiple reviewers requested additional theoretical motivation for our approach. In response, we have enhanced the justification for the clean vs. noisy conditioning paradigm (Figure 2 and Figure 3) and expanded discussions that provide deeper insights into the underlying principles of our method in Section 3.2 "Existing Issues With Noisy Conditional Diffusion Models."
> While these justifications are not purely theoretical, we believe they help to better contextualize our model relative to state-of-the-art methods and provide clearer insights into our method's design.
>
> > I have only one minor requested change. In section 3.3, the authors mention that they apply a mask $x_{supervised} = x \odot m$ and $x_{unavailable} = x \odot (1-m)$. Here, I am supposing $C = A \odot B \iff C_{ij} = A_{ij}B_{ij}$, but I would like the authors to: 1) confirm this, 2) include, in the main text, a definition for this operation.
>
> 1) The reviewer's interpretation is correct. The notation $A \odot B$ represents the Hadamard product (element-wise multiplication), defined as $(A \odot B)\_{i,j} = A\_{i,j} \times B\_{i,j}$.
> 2) We have included this explicit definition in the main text to ensure clarity and mathematical precision.

---

### Author Response · Authors · 2025-07-09
**Global Answer**

We thank the reviewers for their constructive feedback and interest in our work.
We have added additional experiments and discussions as proposed:

Changelog:
- New Figure 2, Figure 3, and update of "3.2 Existing Issues With Noisy Conditional Diffusion Models"
    - Requested by Reviewer Uodi and for Reviewer BEHa, to better motivate the clean vs. noisy conditioning approach
- Updated "Figure 4: Training procedure of MDD" and "3.3 Noise Modeling for Semi-Supervised Multi-Domain Translation"
    - Requested by Reviewers uQXo and Uodi, to clarify multiple aspects of the training process and better explain the different $\phi$ strategies for deterministic or generative settings
- Updated "4.3 BL3NDT Results / Bridge Translation"
    - Requested by Reviewer uQXo, to provide a more thorough discussion of BL3NDT Bridge results
- Updated "5 Limitations and Future Directions"
    - Requested by Reviewer Uodi, to provide further discussion and suggestions on future directions to improve MDD scalability
- Updated "A.4 Model Architecture and Adaptation to Multiple Domains"
    - Requested by Reviewer Uodi, to better explain the architectural details of the models used (MDD, MSDM, and UMM-CSGM)
- New Section "B.3 Sampling Strategy Impact on Diversity"
    - Requested by Reviewers uQXo and Uodi, to better demonstrate the impact of different strategies for generative or deterministic settings on CelebAHQ
- New Section "B.4 Sensitivity to Noise Schedulers"
    - Requested by Reviewer uQXo, to further validate that MDD can extend to different diffusion schedulers
- New Section "B.5 MDD Computational Analysis"
    - Requested by Reviewer uQXo, to compare the inference cost of MDD in different domain settings against state-of-the-art methods

---

### Decision · Action_Editor_RJpR · 2025-08-29

**Recommendation:** Accept as is

**Additional Comments:**

This paper proposes an efficient, flexible, multi-domain translation method that does not require separate models for each configuration. Reviewers raised concerns about the scalability of the fusion strategy and the rationale for selecting clean versus noisy conditions. They also noted the need to add examples of noise masking for missing views, clarify its usage in the algorithm, address error propagation and conflicting factors in the BL3NDT bridge configuration, and explain the sensitivity of noise scheduling. The authors addressed these concerns and made the necessary revisions. These responses and revisions resolved the concerns. Finally, the reviewers decided to accept the paper, and the AE agreed.

**Audience:**

Yes

**Audience Explanation:**

The core method of this research is the diffusion model, which is one of the most prominent machine learning techniques currently receiving significant attention. Furthermore, it can process numerous modalities without substantially increasing the size of the model, which enables the integration of information from different modalities. Additionally, the proposed method allows for semi-supervised domain translation, which mitigates the inability to acquire data in situations where data collection is difficult. These characteristics address issues that may arise in real-world scenarios and are expected to be of interest to a wide range of audiences in TMLR.

**Claims And Evidence:**

Yes

**Claims Explanation:**

This paper presents an efficient, flexible, multi-domain translation method that eliminates the need for separate models for each specific configuration. It features the reconstruction of missing views for new data points and supports semi-supervised learning with arbitrary supervised information configurations.

Extensive experiments on three datasets (BL3NDT, CelebAMask-HQ, and BraTS 2020) demonstrate that the method's performance outperforms adaptive and state-of-the-art baselines.

The authors addressed the reviewers' concerns and revised the paper accordingly. These responses and revisions resolved the reviewers' concerns, and the paper's claims are now supported by clear evidence.